# MassCube improves accuracy for metabolomics data processing from raw files to phenotype classifiers

Huaxu Yu [1], Jun Ding[2], Tong Shen [1], Min Liu [1], Yuanyue Li [1] & Oliver Fiehn [1] ✉

Nontargeted peak detection in LC-MS-based metabolomics must become robust and benchmarked. We present MassCube, a Python-based open-source framework for MS data processing that we systematically benchmark against other algorithms and different types of input data. From raw data, peaks are detected by constructing mass traces through signal clustering and Gaussian-filter assisted edge detection. Peaks are then grouped for adduct and in-source fragment detection, and compounds are annotated by both identity- and fuzzy searches. Final data tables undergo quality controls and can be used for metabolome-informed phenotype prediction. Peak detection in MassCube achieves 100% signal coverage with comprehensive reporting of chromatographic metadata for quality assurance. MassCube outperforms MS-DIAL, MZmine3 or XCMS for speed, isomer detection, and accuracy. It supports diverse numerical routines for MS data analysis while maintaining efficiency, capable for handling 105 GB of Astral MS data on a laptop within 64 min, while other programs took 8–24 times longer. MassCube automatically detected age, sex and regional differences when applied to the *Metabolome Atlas of the Aging Mouse Brain* data despite batch effects. MassCube is available at https://github.com/huaxuyu/masscube for direct use or implementation into larger applications in omics or biomedical research.

MS-based nontargeted chemical analysis has rapidly advanced for lipidomics, exposome analyses and metabolomics in biomedical and environmental research[1-4]. Today, studies involve thousands of data files across different assays, often involving many laboratories. The UK Biobank collected 118,461 human plasma samples that await nontargeted LC-MS analyses[5]. Meanwhile, new generations of mass spectrometers such as the Orbitrap Astral or timsTOF instruments have led to eightfold increases in raw data file sizes, including a much higher volume of tandem mass spectra (MS/MS). This growth in size of both biological studies and MS data files requires more accurate, robust, and efficient software for data management and processing.

Nontargeted LC-MS data processing must report all detected chemicals with high sensitivity and robustness. Existing software has often not been thoroughly validated and benchmarked for data processing efficiency and accuracy of final reports that start from raw data imports and management to feature detection, recognition of genuine chromatographic peaks in lieu of false positives, adduct grouping including in-source fragments (ISF), chromatographic alignments, exhaustive structural annotations to statistical analysis and visualization. For example, XCMS[6], MS-DIAL[7], MZmine[8], El-MAVEN[9], OpenMS[10], and TidyMass[11] do not directly annotate ISFs and often report many false positive features that cannot be validated as true chromatographic peaks. Software deficiencies has led to developments of

[1]West Coast Metabolomics Center, University of California Davis, Davis, CA, USA. [2]China CAS Key Laboratory of Plant Germplasm Enhancement and Specialty Agriculture, Wuhan Botanical Garden, Chinese Academy of Sciences, Wuhan, PR China. ✉e-mail: ofiehn@ucdavis.edu

ancillary software packages to rectify problems and missing steps in nontargeted study analyses, such as CAMERA[12] and NetID[13] for feature grouping, GNPS[14], SIRIUS[15], and BUDDY[16] for compound annotation, NOREVA[17] and SERRF[18] for normalization, and MetaboAnalyst[19] for statistical analysis.

Two primary issues remained thorny challenges. First, feature detection, the basics for MS data processing, proved difficult to balance accuracy and speed. Classic software has focused on evaluating the rate of change in *m/z* abundance across a chromatographic peak for feature detection. However, features in LC-MS experiments can be noisy and often lack baseline separation. Rate-of-change approaches are inherently limited in robustness and are highly sensitive to noise. Consequently, automated feature detection results require extensive inspection by experienced analytical chemists, which becomes increasingly difficult as the size and number of biological studies increase. Besides accuracy, processing speed is critical for enabling data curation and downstream data evaluation, including quality controls and initial characterization of biological variance in the data. While the recent development of *Flash Entropy Search*[20] enables ultrafast and comprehensive MS/MS investigations, chromatographic information is often overlooked due to the limitations of current feature detection algorithms in handling large-scale data processing tasks. The second major issue with MS data processing software concerns the evolution and integration of algorithms. While many method developments have achieved breakthroughs in addressing specific data processing problems, incorporating such advancements into the entire pipeline for routine research use is often difficult and time-consuming. Challenges arise due to incompatible data formats, naming conventions, or rigid software structures. Most popular metabolomics software lacks the flexibility needed for user-driven extension and modification tailored to application-oriented data processing.

We therefore present MassCube as an open-source computing library and framework for MS data processing built in Python. MassCube supports comprehensive functionalities and workflows designed for versatile data processing tasks. Inspired by the approach used in *centWave*[21], feature detection in MassCube employs a signal-clustering strategy coupled with Gaussian filter-assisted edge detection algorithm. Compared to MS-DIAL, MZmine3, and XCMS, MassCube demonstrated superior feature detection coverage, accuracy, and speed across both synthetic and experimental MS data. We chose Python for MassCube to leverage the advancements in array programming and to take advantage of Python's extensive ecosystem of numerical libraries, its large user base, machine learning frameworks, and its balance between performance and ease of use. The object-oriented and modular architecture of MassCube facilitates the rapid implementation of algorithms contributed by the community, such as *Flash Entropy Search* for fast and advanced MS/MS matching.

## Results

### Software functionality

MassCube comprises 16 modules to handle all data processing tasks for MS-based nontargeted chemical analysis (Fig. 1a). The workflow completes importing files, detecting all feature, defining peaks including adducts and ISFs, normalizing retention times and intensities, annotating compounds, performing statistics, visualization, and exporting clean results. Parallel computation enables efficient handling of large-scale data files with minimal memory overhead, allowing complex analyses to be completed even on personal laptops. MassCube is compatible with Windows, macOS, and Linux operating systems to be usable across platforms and frameworks. To support with the FAIR principles, MassCube provides a standardized metadata tracking system that records data processing steps, parameters and version of dependencies, ensuring that each data analysis can be accurately tracked and reproduced.

By integrating processing modules, MassCube offers a choice of different workflows for users without coding skills (Supplementary Fig. 1). By providing fully automated end-to-end data processing, users without substantial resources and technical training may not only obtain clean results but also overcome challenges in quality control, workflow construction or parameter selection. For advanced users and database programmers, MassCube is a ready-to-expand software platform. The modular, object-oriented design of MassCube

**a** MassCube functionalities

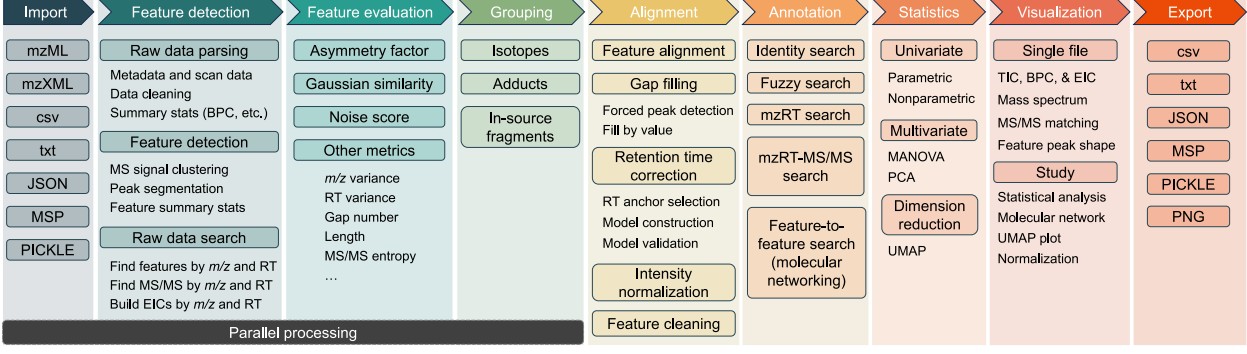

**b** MassCube architecture for flexible extension

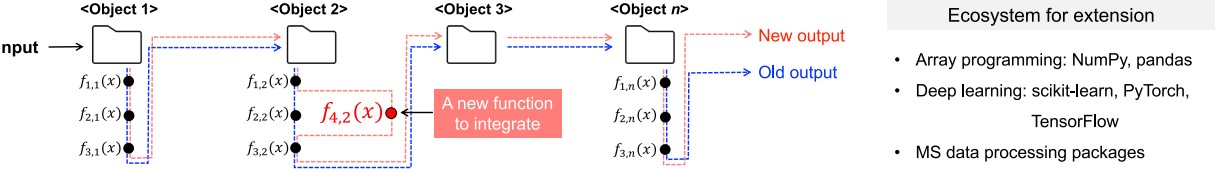

**Fig. 1 | MassCube design. a** Main functionalities in MassCube handle data import, raw data processing, feature detection and segmentation, peak grouping, peak alignment, compound annotation, statistical analysis, data visualization, and data export. **b** The MassCube Python framework supports object-oriented programming and flexible integration of algorithms.

establishes the fundamental structures for MS data, with optimizations for efficient array programming and scalability (Fig. 1b). This design facilitates the flexible integration of advanced algorithms into a complete data processing pipeline for continuous improvement. For example, after publishing *Flash Entropy Search* in 2023, MassCube easily integrated this algorithm including multi-modal (identity search and fuzzy search) MS/MS matching. In comparison to a combination of standalone software packages, the integrated MassCube workflow design eliminates issues arising from incompatible data formats or manual data format adjustments when transferring intermediate results between different modules.

MassCube feature detection empowers accurate feature evaluation, grouping and annotation by clustering all detected MS signals to unique ions (Fig. 2a). Using mass resolution parameter settings, MassCube defines identical $m/z$ values across continuous MS1 scans without imposing requirements on peak shape, signal fluctuation, or scan number, thereby minimizing empirical biases. Merging $m/z$ signals ensures that every detected MS1 signal is assigned to a feature for 100% coverage, maximizing information gain. In difference to rate-of-change peak picking, a second advancement in MassCube is to segment features via a Gaussian filter-assisted edge detection algorithm. This method distinguishes segments from noise or from isomeric peaks. Background noise, characterized by severe signal fluctuation, is ignored by the segmentation process to prevent false positives during feature detection. Third, segmentation allows MS1 signals to be differentiated into distinct chromatographic peaks, improving detection of isomers. Importantly, the Gaussian filter, as a smoothing algorithm, is only utilized for robust peak edge detection, but not for calculating peak areas and heights. Depending on the type and parameters of algorithms, smoothing may introduce substantial changes in data structures and introduce bias. Therefore, MassCube uses the raw data instead of smoothed data for the final reports.

## Benchmarking using synthetic MS data

The expected true positive rate for feature detection is determined by three peak parameters: signal-to-noise ratio (S/N), peak resolution, and the peak intensity ratio relative to adjacent peaks (Fig. 2b). Under conditions of low S/N, low peak resolution, and high intensity ratios, defining the true presence of peaks can be challenging, even for experienced analytical chemists (Fig. 2c). These challenges are further exacerbated if S/N threshold was lowered to detect low-abundant compounds, or if signal fluctuation across peaks increases. The key to successful peak detection is to balance the trade-off between sensitivity and robustness. Ideally, the algorithm must report exactly one feature for a single peak and two features for a double peak, possibly even a shoulder peak (a partially resolved peak pair where one peak has lower intensity than the other). This definition becomes more challenging as the signal fluctuation increases. If the algorithm is overly sensitive, a single peak may be incorrectly split into multiple features; conversely, if the algorithm is too insensitive, isobaric species may not be properly distinguished.

We optimized the performance of MassCube's peak detection by tuning algorithm components critical to balancing the sensitivity-robustness tradeoff. Specifically, we focused on optimizing the segmentation algorithm by adjusting two key parameters: the sigma value ($\int$) in the Gaussian filter function, which controls noise tolerance, and the peak prominence ratio, which determines sensitivity to local minima (see "**Methods**"). For comprehensive evaluation, we designed a synthetic dataset by varying these three peak parameters, generating a total of 110,000 distinct MS signals for single peaks and another 110,000 double-peak signals. These simulations covered peak detection scenarios ranging from easy to challenging, including both single and double peaks for isomer detections. The benefit in using synthetic data is to define true positive peaks beforehand, instead of only relying on subjective judgments by analytical

chemists. When $\int$ and peak prominence ratio are high, the algorithm is very robust to noise and accurate to detect single peaks, but at the expense of reduced sensitivity in distinguishing double peaks (Fig. 2d). To improve double-peak accuracy, a moderate selection of $\int$ and prominence ratio is needed, ensuring that the algorithm is neither too sensitive nor too insensitive. Because the number of double-peaks and single-peaks in experimental LC-MS data is sample- and method-dependent, MassCube's configuration for peak detections was optimized to an overall best average accuracy. This process achieved an average accuracy of 96.4% with optimal settings of $\int = 1.2$ and prominence ratio = 0.1.

Next, to benchmark MassCube's peak detection against other software, we generated a synthetic mzML file from a negative mode electrospray QTOF MS dataset of human urine, where 13,500 true single peaks and 13,500 true double peaks were inserted at $m/z > 1500$ Da to insure that test signals were not interfering with experimental data. All synthetic signals were modeled with >10 scans, varying Gaussian noise fluctuations from 0 to 10% and peak height ratios of double peaks from 1–5, with peak resolution varying from 1 to 2. We then used this synthetic data to statistically compare the performance of MassCube against MS-DIAL 4.9, MZmine 3.90, and *xcms* R package 4.0.0 (Fig. 2e). For double-peak accuracy under six noise scores ranging from 0 to 10%, MassCube achieved the highest mean accuracy of 95.2%, significantly higher than MZmine3 (mean = 87.0%, paired *t*-test $p = 0.0011$) and *xcms* (mean = 76.0%, $p = 0.0010$), but not significantly higher than MS-DIAL (mean = 94.3%, $p = 0.27$). Mass-Cube also attained the second-highest mean accuracy for single peaks at 97.8%, significantly higher than MS-DIAL and MZmine3 yet with no significant difference compared to *xcms* (mean = 98.8%, $p = 0.58$). Overall, MassCube achieved the highest average accuracy of 96.5%, significantly outperforming MZmine3 (mean = 88.4%, $p = 0.0029$), *xcms* (mean = 87.4%, $p = 0.016$), and MS-DIAL (mean = 85.4%, $p = 0.0065$). Using this dataset, peak detection in MassCube was found as the most robust and sensitive software in comparison to MS-DIAL, MZmine3, and *xcms*.

## Benchmarking using experimental MS data

Experimental data may differ from synthetic data in unexpected ways that are hard to simulate, such as dips, tailing, insufficient scan numbers (including single scan ion detections), raising baselines or missing signals.

We started by distinguishing a feature in LC-MS datasets from genuine chromatographic peaks. A feature in MassCube is every detected unique $m/z$ signal after clustering. A peak in MassCube is defined as a feature that is segmented by the Gaussian filters into at least one segment that is different from the total chromatographic run time with a minimum of 5 scans. All peaks have associated quality metadata to enable users to prioritize peaks in final data reports, including peak asymmetry factor, Gaussian similarity, and noise scores (Supplementary Fig. 2). While peak detections by segmentation allow for missing values, such missed signals will contribute to poor quality metadata assignments. Using 200 files from 41 studies downloaded from MetaboLights[22], we then profiled the metadata distribution of 1,442,223 peaks detected with ε5 scans per extracted ion chromatogram (Supplementary Fig. 2). For example, a perfectly symmetric peak is expected with peak asymmetry factor of 1, while tailing or fronting peaks differ from that value. As expected, the number of peaks exhibiting fronting and tailing were similar. A Gaussian-shaped peak has a dot-product of 1 to the fitted Gaussian distribution. Over 50% of all peaks had a Gaussian similarity over 0.84, while the remaining features showed a nearly uniform distribution of Gaussian similarity between 0.2 and 0.8, indicating the presence of noise. The median noise score was 0.43, suggesting that signal fluctuation is common in untargeted chemical analysis. Only 25% of all peaks showed noise scores <0.20, indicating low signal fluctuation.

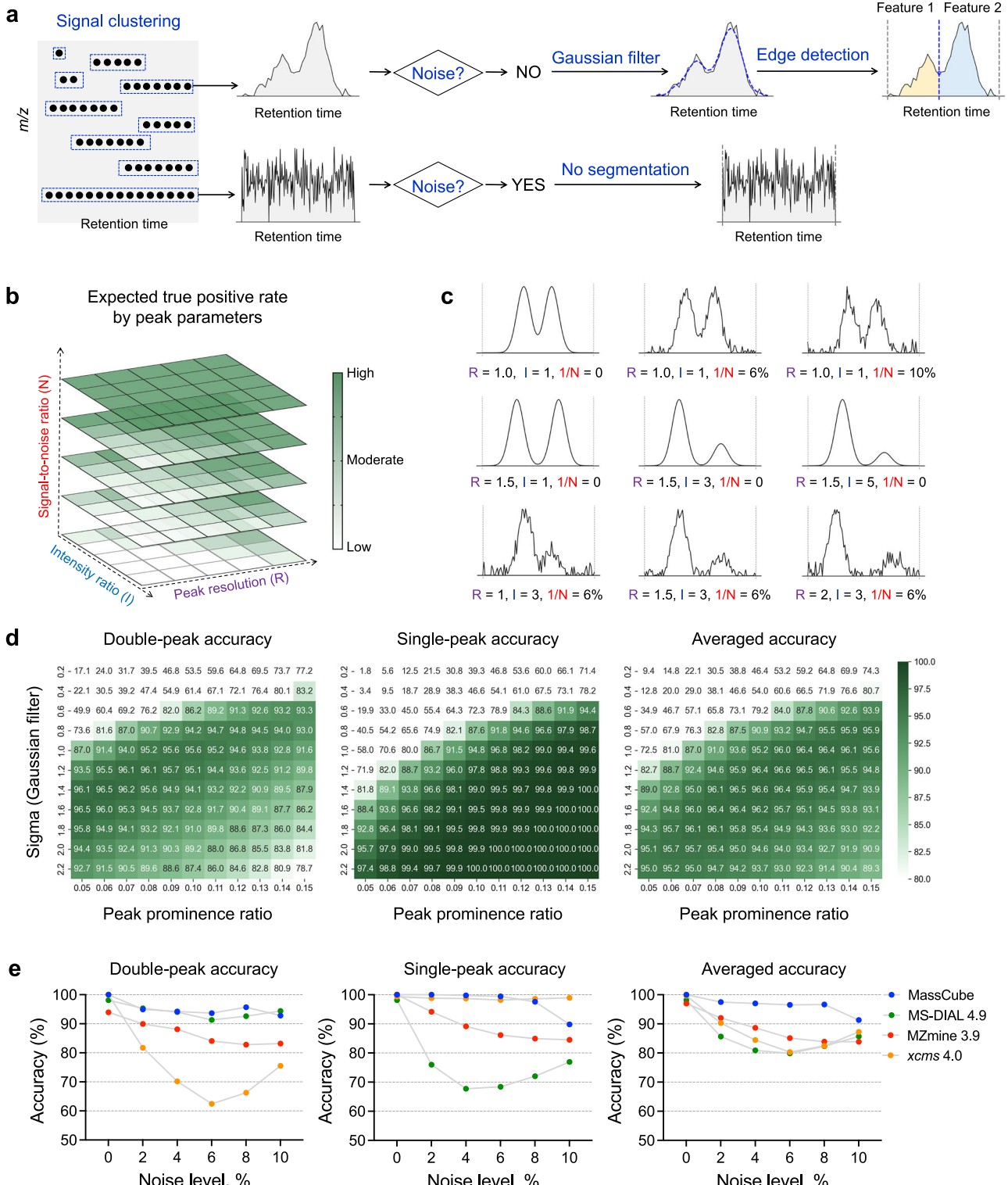

**Fig. 2 | Benchmarking the MassCube workflow for peak detection using simulated MS data. a** MassCube starts by clustering MS1 signals, determines noise, and segments features into chromatographic peaks. **b** Schematic representation of the impact of peak parameters resolution, intensity ratio, and signal-to-noise ratio on peak detections. **c** Visual examples for simulated double peaks under different peak parameters. **d** heatmaps of true-positive peak detections for 220,000 simulated peaks for optimizing parameters during MassCube algorithm development. The averaged accuracy (right panel) determined the best balance between sensitivity and robustness. **e** Benchmarking double-peak, single-peak, and averaged accuracy for 27,000 simulated peaks for MassCube, MS-DIAL, MZmine3, and *xcms* software.

Metadata distributions showed that hard thresholds for peak quality cannot be ascertained in a straightforward way. Yet, users can scrutinize and curtail peak detection using the combined impact of these quality scores from final MassCube reports.

We therefore benchmarked MassCube's peak detection against leading software using diverse experimental MS data. To ensure fair, objective and reproducible comparison, we implemented three key approaches (see "software and parameters" in "Methods" section).

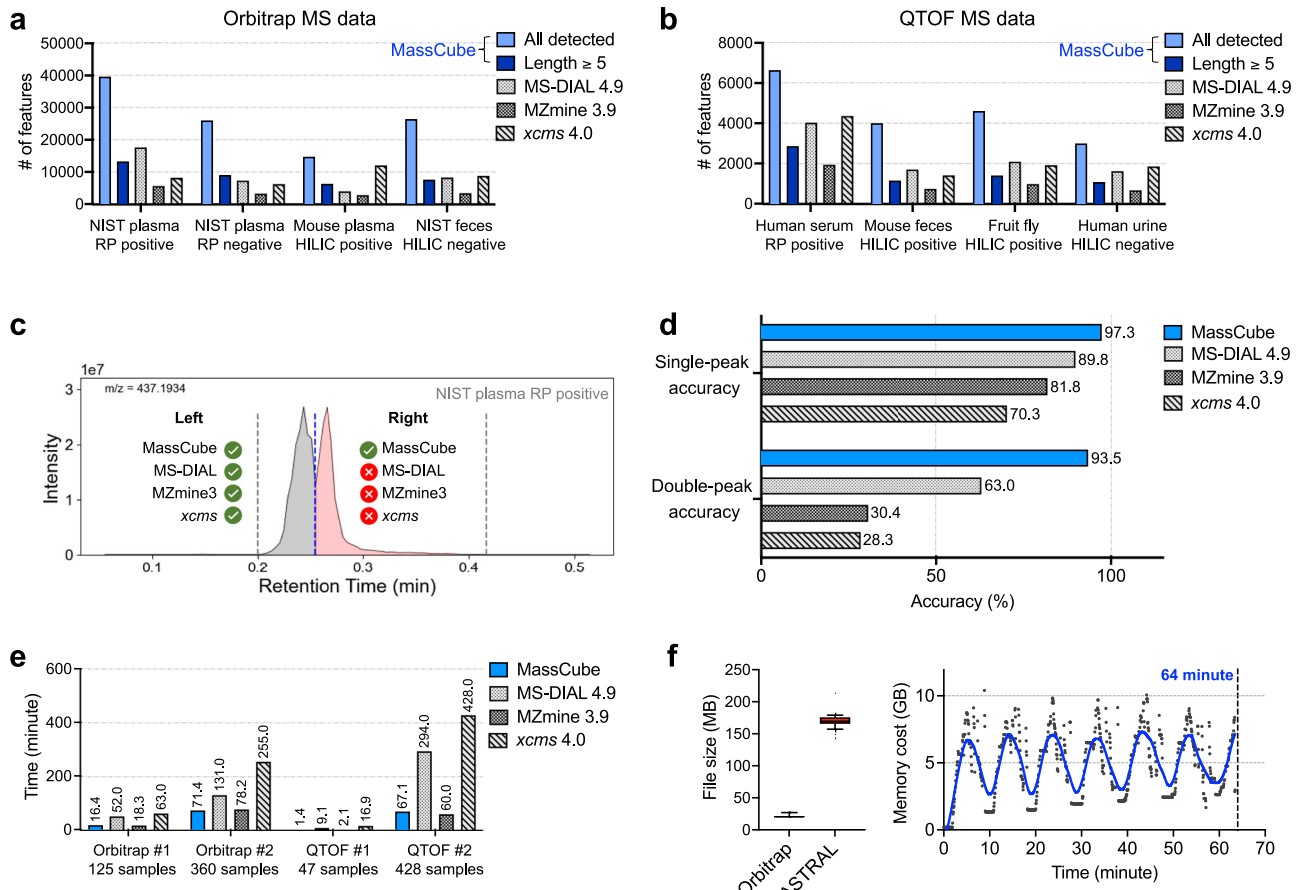

**Fig. 3 | Benchmarking MassCube peak reporting results against MS-DIAL, MZmine and *xcms* using experimental MS data.** Eight ThermoFisher Orbitrap QExactive or Bruker QTOF Impact2 LC-MS/MS data files were used for metabolomics and lipidomics analyses of NIST SRM1950 human plasma, human serum, mouse plasma, NIST RGTM 10162 fecal matter, whole body of fruit flies and human urine samples. MS-DIAL vs.4.9 was used by default settings, *xcms* vs.4.0 with default settings 5–60 s peak widths, MZmine3 vs. 3.90 with default settings ε5 non-zero consecutive scans, and MassCube with no restrictions (all features) or peaks with ε5 scans per segment. **a** Comparisons of the number of reported peaks for Orbitrap data. Grey: all detected features by MassCube; blue: MassCube reported peaks with ε5 scan points. Other software: MS-DIAL (green), xcms (red), MZmine3 (orange). **b** Comparisons of the number of reported peaks for QTOF data. **c** Example of a chromatographically partly resolved accurate mass pair in NIST SRM1950 plasma detected by MassCube but not detected by other software. **d** Benchmarking true

positive peak picking. 722 features from QTOF and Orbitrap QC results were randomly selected and manually verified by visual inspection using extracted ion chromatograms. Double-peak: 46 partially resolved accurate mass peaks. Single-peak: 676 chromatographic single-peak accurate mass features. **e** Benchmarking data processing speed on four public datasets (plasma, urine, stool, plant leaves) with 47–428 sample files using single threading. **f** MassCube performance for very large file sizes. Comparison of file size for 636 human plasma samples of Alzheimer's Disease patients acquired on an Orbitrap Exploris 240 (median 20.4 MB per sample) and an Orbitrap Astral mass spectrometer (median 169.5 MB per sample), using a MacBook M3 Pro laptop. Parallel processing was enabled with a batch size of 100, with a total processing time of 18 min for the Orbitrap Exploris 240 data and 64 min for the Orbitrap Astral data. Right panel: memory usage for Orbitrap Astral data processing; the dashed line represents the smoothed curve. Whiskers: 5–95 percentile; box: 25–75 percentile with the center indicating median.

First, standardized key parameters were used in a consistent way for all tested software, including intensity cutoffs and mass tolerances. Second, each software used its recommended default values for all unique parameters that were not shared with other software. Because different algorithms operate under distinct mathematical assumptions, using recommended parameter settings ensure a fair use of software as intended by the software developers. Third, diverse datasets were selected to prevent bias or overfitting towards any specific software. Here, datasets ensured (a) matrix independence by selecting biologically relevant (plasma, human, mouse) and complex (fecal) matrices, (b) reusability by utilizing NIST reference materials, (c) size independence by including both small- and large-scale studies, and (d) instrument independence by testing both Orbitrap and QTOF datasets.

Therefore, a total of eight LC-MS data files from samples including NIST SRM 1950 plasma, mouse plasma, NIST Human Fecal Material RGTM 10162, human serum, mouse feces, whole fruit fly (*D. melanogaster* strains), and human urine data, acquired on Orbitrap MS and QTOF MS with different ion mode (Fig. 3a, b). LC-MS runs were

acquired under data-dependent MS/MS conditions, but MS/MS signals were not used in peak detection. While for Orbitrap data >10,000 features were detected in all data files, only a fraction of these features was present in peak segments with ε 5 scans (Fig. 3a). Similarly, while for QTOF data >2000 features were detected in all data files, we found a similar ratio as in Orbitrap data files for the ratio of the total number of detected features over peaks with length ε 5 scan as (Fig. 3b). Overall, 35% of all peaks showed acceptable scan lengths, suggesting that many features in LC-MS files are spurious and cannot be associated with high quality chromatographic peak metadata defined by peak asymmetry, Gaussian similarity and noise scores (Supplementary Fig. 2). This observation also justifies a distinction between LC-MS 'features' and chromatographic 'peaks'. While peaks can be used for defining genuine signals that may be attributed to the presence of specific chemicals, other features may occur due to background contaminations that are caused by inconsistent electrospray patterns, or by very low abundant metabolite signals that do not reach a threshold to be defined as peaks. Interestingly, up to 28% of

these spurious features with low scan numbers were associated with MS/MS data in data-dependent LC-MS/MS acquisitions, because >75% of them were present at peak intensities that were substantially higher than the background noise for both Orbitrap and QTOF data. Mass-Cube demonstrated the ability to detect all features (including spurious signals), attributed to its ion clustering-based feature detection strategy that ensures no signal loss. When defining peaks as signals with ε 5 scan lengths, MassCube reported a similar number of peaks as other software. However, the accuracy of peak reporting can be different between software packages and parameters. A true single peak might be split into multiple reported features, while a true double peak might be reported as a single feature, resulting in similar numbers of detected peaks. Figure 3c illustrates an example where a true peak pair in an experimental data file was not distinguished by either MS-DIAL, MZmine3 or *xcms*, but accurately detected by MassCube, underscoring the accuracy of MassCube's segmentation algorithm. To comprehensively probe the accuracy of the four software tools, we randomly selected 722 locally extracted ion chromatograms with retention time window ±0.2 min and manually labeled the number of peaks in the range (see "**Data availability**" for extracted ion chromatograms). The labeled data were compared with the results from MassCube, MS-DIAL, MZmine3, and *xcms*. MassCube achieved the highest accuracy for double peaks (93.5%) and single peaks (97.3%), outperforming the other three software tools (Fig. 3d). These results suggest that MassCube's feature detection algorithm closely approximates the results obtained by experienced analytical chemists, demonstrating the best performance for scaling human-level data processing into automated, large-scale workflows.

Apart from accuracy of peak reporting, MS-based chemical analysis must efficiently process large numbers of files, for example, in human cohort study samples. Often, LC-MS data processing software crashes for large file numbers. MassCube was designed to maximize efficiency by incorporating Numpy-based array programming and parallel processing. In benchmarking tests across four datasets with varying sample sizes using single-threading (Fig. 3e), MassCube demonstrated substantially faster feature detection speeds compared to *xcms* (average 6.5-fold improvement) and MS-DIAL (average fourfold improvement), and slightly faster speeds compared to MZmine3 (average 1.15-fold improvement). Despite Python's inherently slower execution speed compared to Java (used by MZmine3), these results underscore Mass-Cube's low computational cost. We further demonstrated MassCube's capability by processing a large-scale Orbitrap Astral MS dataset consisting of 636 files with a total size of 105 GB (Fig. 3f). On a MacBook M3 Pro laptop equipped with 36 GB of memory and 12 cores, MassCube completed the parallel data processing for all files in 64 min, with peak memory usage reaching only 10.4 GB. This low memory usage enables a standard laptop to efficiently handle next-generation MS data with eightfold larger sizes than classic Orbitrap instruments, paving the way for the processing of increasingly complex datasets.

## Biological application by data re-analysis

We demonstrated the capability and performance of MassCube by reanalyzing the *Metabolome Atlas of the Aging Mouse Brain* dataset[23]. Data encompassed a total of 702 samples with positive and negative ESI for HILIC-based metabolomics and RPLC-based lipidomics across 80 sample groups (representing ten regions, two sexes, and four age groups). MassCube handled the entire process for comprehensive data analysis (Fig. 1 and Supplementary Fig. 1). In particular, MassCube automatically resolves isomer peaks (Fig. 4a, b), groups adducts and ISFs (Fig. 4c), aligns all peaks across chromatograms and performs automatic compound annotations, including chemical classifications for unknowns. MassCube also performs data normalization to reduce retention time shift (Supplementary Fig. 3) and batch effects (Supplementary Fig. 4), revealing sex-specific metabolome differences in brain regions that were not reported in the original publication[23].

With MassCube's accurate peak detection algorithm, isomeric peaks can be effectively resolved and compared across sample groups. We demonstrated its ability to identify previously unresolved isomeric lipids, such as phosphatidylcholine (PC 36:2, Fig. 4a) and hexosylceramide (HexCer 41:2;3 O, Fig. 4b), which were not distinguished by MS-DIAL 4.90. By analyzing mouse brain data from adolescence and early adulthood groups, we found clear differences in the statistical significance between the left and right peaks in both examples, highlighting the importance of precise peak detection and resolution. For instance, the right peak of PC 36:2 (Fig. 4a) showed much better significance ($p = 5.1 \times 10^{-5}$) compared to the right left ($p = 1.1 \times 10^{-3}$), showing MassCube's ability to distinguish such isomers.

Retention time drift is a critical challenge in large-scale MS analysis or collaborative data acquisition across laboratories. MassCube uses an automatic detection of unique model peaks that can be used to anchor chromatograms across substantial retention time shifts. This approach is different to retention time corrections that use internal reference standards (or fixed sets of matrix compounds), and it is also different from classic approaches that attempted to align chromatograms via overall similarity. The classic approach fails if studies incorporate samples with very distinct metabolome profiles, such as kidney tissues and urine samples. Similarly, brain sections in the mouse metabolome atlas were also quite different in total metabolome profiles, specifically across regions (Fig. 4b). After depicting unique model peaks (defined by unique *m/z* segments with high peak quality and intensity) in QC pools samples, MassCube uses these model peaks via linear interpolation, specifically designed to handle nonlinear RT shifts in LC-MS experiments (Supplementary Fig. 3a, b). These anchors are then split into training and testing sets for retention time correction and validation, respectively. The correction algorithm effectively reduced retention time shifts across all four analytical modes, demonstrating better performance in reverse phase mode by reducing retention time shifts in the test sets from 31.0 s to 6.0 s in positive mode and from 40.5 s to 1.0 s in negative mode (Supplementary Fig. 3c–e). Data normalization was performed by automatically acquiring timestamps from the raw data to define the acquisition order and applying locally weighted scatterplot smoothing algorithm to correct for systematic signal intensity drifts (Supplementary Fig. 4).

MassCube automatically searches all MS/MS spectra against MassBank of North America[24] and the MS-DIAL metabolomics MSP spectral kit[7], but it can incorporate other MSP libraries as well. For the re-analysis of the mouse brain atlas data, we also incorporated the licensed NIST23 library. With a flash entropy search similarity cutoff of 0.7, MassCube annotated a total of 1710 unique compounds, separated into 533, 286, 711, and 613 unique compounds from 7888, 7049, 12,270, and 13,984 peak groups in HILIC positive and negative, RPLC positive and negative, respectively. Hence, a substantial number of peak groups remained unidentified. Beyond classic identity search (using precursor *m/z* and MS/MS matching), MassCube now fully supports MS/MS fuzzy searches (also known as molecular networking[14]) for chemical classification of unknown compounds, combining open search, neutral loss search, and hybrid search[20,25]. Classifying unknown compounds enables class enrichment statistics[26], and it also improved biological interpretation to generate hypotheses. This fuzzy search extended the number of chemically classified compounds two-threefold across the four assays[27] (Supplementary Figs. 5 and 6). Without precursor restrictions in fuzzy search, the search space increases more than 6000-fold (Supplementary Fig. 7). Despite this massive increase, total search time was only <1% longer relative to the total data processing time. When stratified into chemical classes by fuzzy search, age- and regional-specific metabolic differences became statistically significant for several compound classes such as hexosylceramides and sterols that were not previously published[23] (Supplementary Fig. 8). To

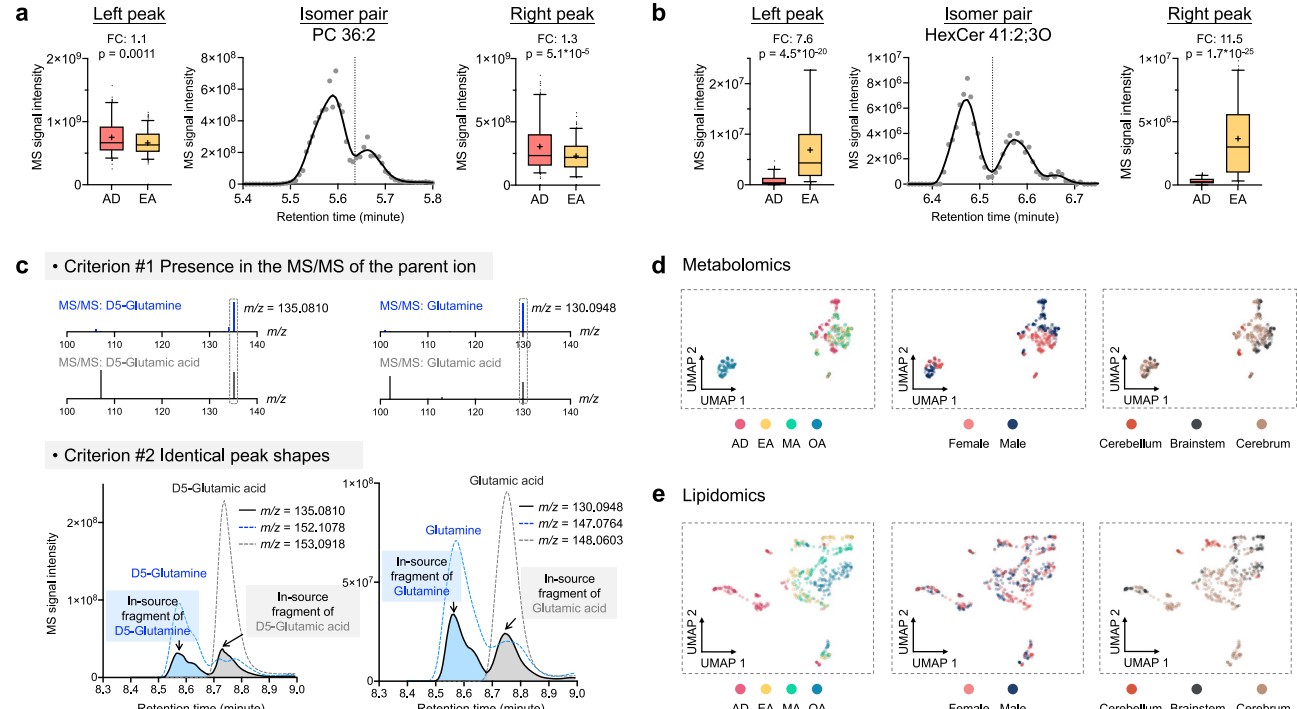

**Fig. 4 | Validating MassCube reports by re-analyzing raw data of the *Metabolome Atlas of the Aging Mouse Brain*. a, b** Examples of isomer pairs including (**a**) phosphatidylcholine PC 36:2 and (**b**) hexosylceramide HexCer 41:2;3 O detected by MassCube but not resolved by MS-DIAL 4.90. Box plots illustrate the difference of statistical significance of lipid quantity between the left and right peaks when comparing adolescence (AD) and early adulthood (EA) groups with 160 biological samples in each group. FC fold change; *p*: two-tailed *t*-test *p*-value with unequal variance; whiskers: 5–95 percentile; box: 25–75 percentile with the center indicating median. **c** Automated identification of in-source fragments in MassCube. Two criteria are combined, exemplified here for the internal standards D5-glutamine (*m/z* 152.1078) and D5-glutamic acid (*m/z* 153.0918) and their corresponding endogenous metabolites. *Top panels* (criterion #1): Presence of in-source fragment *m/z* 135.081 (for the deuterated internal standards) and 130.0948 (for the endogenous metabolites) in the MS/MS spectra of the corresponding parent [M + H]⁺ ions. *Lower panels* (criterion #2): identical peak shapes of in-source and parent ion *m/z* values

for loss of NH₃ (D5-glutamine and endogenous glutamine) or loss of H₂O (D5-glutamate and endogenous glutamate) to the same fragment ion *m/z* 135.081 (for D5-labeled internal standards) or m/z 130.0948 (for the endogenous metabolites). Peak shape identities were calculated scan-by-scan in MS1 profiles by Pearson's product-moment correlation coefficient. **d, e** Uniform manifold approximation and projection (UMAP) clusters of 702 sample files reprocessed by MassCube for (**d**) mouse brain metabolomics and (**e**) mouse brain lipidomics. Coloring the UMAPs by sample type revealed the age-related, sexual, and regional differences. Left panels in (**d, e**) show age. AD: adolescence, 3 weeks of age; EA early adulthood, 16 weeks of age; MA middle-age, 59 weeks of age; OA old age, 92 weeks. Mid panels in (**d, e**) show sex difference present in metabolomics but not in lipidomics data. Right panels in (**d, e**) show differences between the three major brain parts cerebellum, cerebrum and brainstem. Lipidomics data also revealed a specific region, the olfactory bulb (indicated by an oval), to be highly distinct from other regions in the cerebrum.

properly annotate unknown compounds that are classified by not identified, additional follow-up experiments are needed such as deuterium/hydrogen exchange studies[28]. Next, the re-analysis of the mouse brain atlas data validated how ISFs are confidently annotated in MassCube. Unlike isotopes and adducts, the annotation of ISFs is more challenging because no prior knowledge is available for the compound-dependent *m/z* difference between an ISF and its parent ion[29]. Current MS data processing software tools do not clearly annotate ISFs in the final report, often requiring analyses of correlations across multiple chromatograms and extra software for data curation. MassCube uses a different strategy, using only the available data within each chromatogram, systematically annotating ISFs using two key criteria (Fig. 4c). The first criterion recognizes that in-source fragmentation shares similarities to low-energy MS/MS fragmentations. Here, MassCube checks if a potential ISF ion also appears in the parent MS/MS spectrum. Secondly, MassCube examines the scan-to-scan Pearson correlations between all co-eluting ions because ISFs intensity must directly depend on the intensity of the intact precursor ions. Figure 4c demonstrates the proposed strategy for both internal standards and endogenous metabolites. Our proposed approach correctly annotated two ISFs from two true positive peaks, the internal standards, D₅-glutamine and D₅-glutamic acid. The findings were further verified by correctly

annotating ISFs for both non-labeled endogenous glutamine and glutamic acid. Overall, MassCube detected 2604 ISFs in the mouse dataset, in addition to 6055 alternative adducts that were then combined to unique peak groups. For detected ISFs, 7.5% (194 out of 2604) were classic water losses, and 4.1% (107 out of 2604) were losses of ammonia. Hence, MassCube opens the door for more detailed analyses of the chemical nature of other in-source fragmentations.

To display overall metabolic phenotypes, MassCube generated unified nontargeted results from annotated compounds that were detected in > 80% of all samples. In this way, global differences in the brain metabolome (Fig. 4b) and lipidome (Fig. 4c) were mapped using 431 unique metabolites and 953 unique lipids, respectively. This data reduction strategy enabled focusing on metabolomics differences across three major brain regions including the cerebellum, brainstem, and cerebrum, clustered from ten subregions. Regional tissue differences were evident and aligned with the findings in the original work. Importantly, aging introduced greater variance in the brain metabolome, while sex discriminated tissues across all ages, particularly in the metabolome, though the differences in the lipidome were relatively smaller. Notably, the olfactory bulb, as part of the cerebrum, exhibited distinct lipidomic profiles compared to other cerebrum subregions.

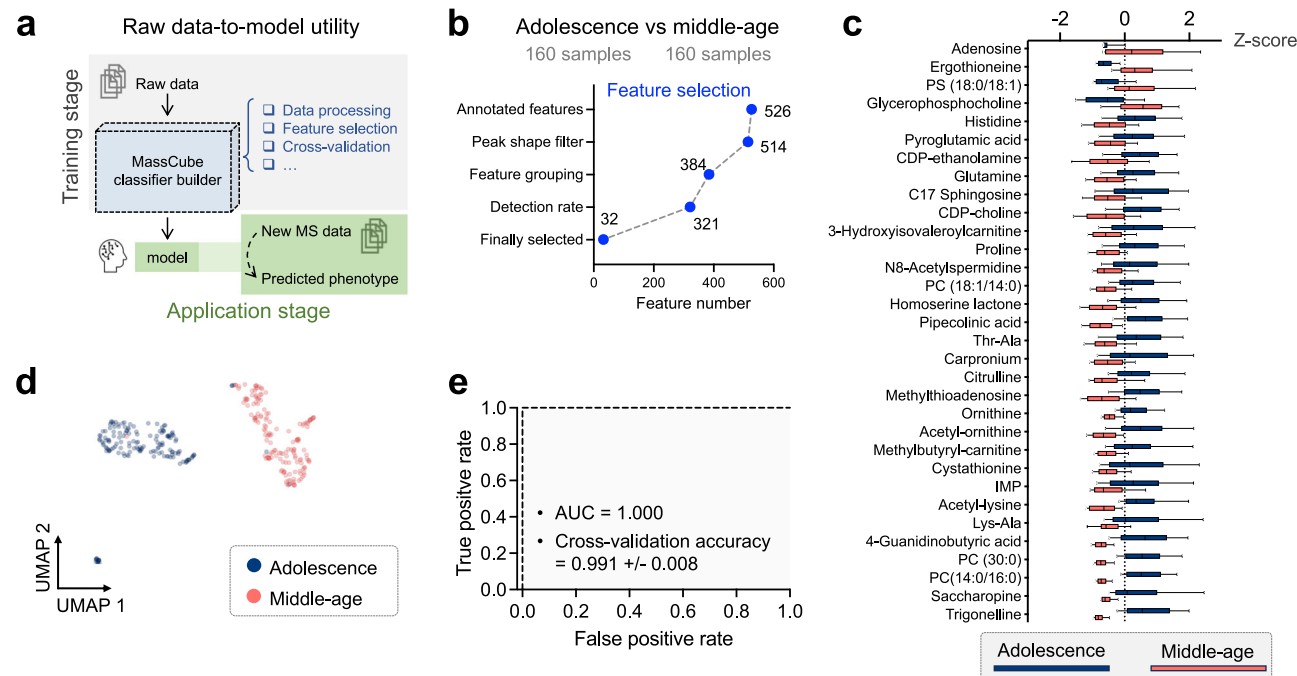

**Fig. 5 | MassCube utility for phenotype classifiers: predicting age-differences across all brain regions, re-using data from the *Metabolome Atlas of the Aging Mouse Brain*. a** Schema for automatic phenotype classifications with two key stages (*i*) the raw data-to-model utility for the training stage, and (*ii*) the raw data-to-phenotype utility for the prediction stage. **b** Querying the phenotype 'age' as discriminant for adolescence and middle-aged mice using 526 named metabolites annotated by *m/z* and MS/MS matching (HILIC, ESI-positive, QExactive, annotation > 0.7 entropy similarity). A random forest classification model was built leading to 32 final metabolites by filtering for MassCube peak metadata and classification strengths. **c** Box and whisker plot of overall metabolite intensities of the 32 selected metabolites between adolescence and middle-aged mouse brain samples with 160 biological samples in each group. Whiskers: 10–90 percentile; box: 25–75 percentile with the center indicating median. **d** UMAP plots depicting the global differences between adolescence and middle-aged mouse brain samples using the selected 32 metabolites. **e** Using the 32-dimensionsional space as classifier leads to an excellent area under the receiver-operator curve (AUC) and fivefold cross-validation accuracy of the model evaluation to distinguish adolescent (3 weeks) from middle-age mice (59 weeks).

## Biological application by machine learning-based phenotype classification

MassCube ranks chemical discriminators between classes of samples. Subsequently, it creates a phenotype classifier model directly from raw MS data, based on the selected discriminators. Python, with its enriched ecosystem for machine learning and artificial intelligence, provides MassCube with capabilities for this task. The MassCube workflow module 'phenotype classifier builder' enables end-to-end construction of a classifier from raw MS data without requiring user expertise for coding skills, knowledge of LC-MS data processing, or machine learning (Fig. 5a). The classifier builder handles routine tasks such as raw data processing, feature selection, model validation, and more, providing users with a preliminary model for sample group prediction. During the prediction stage, users only need to provide the raw LC-MS data, while the MassCube model will automatically locate optimal metabolites used for prediction.

We demonstrated the module by constructing a random forest model to predict subject age using a subset of mouse brain data (160 adolescence and 160 middle-age). After raw data processing, a sequential strategy was employed for selecting features, constructing models, and validating models. Feature selection in MassCube goes beyond conventional statistics-based approaches that focus solely on distinguishing sample groups. MassCube also considers analytical performance factors such as peak shape, feature grouping, and detection rate (Fig. 5b). This strategy ensures that the selected features are analytically reliable, and the constructed model are robust. Yet, results from any single study must be viewed as preliminary outcome because underlying matrix effects may highlight specific metabolites that

are unrelated to biological causes. Out of 526 annotated metabolites, the MassCube workflow selected 32 metabolites with distinct concentrations between adolescence and middle-aged mice (Fig. 5c). This automatic selection concurred with previously reported alterations in the mouse brain metabolome during aging. For instance, adenosine was statistically significantly increased in middle-aged mice compared to adolescence, aligning with results published in the *Metabolome Atlas of the Aging Mouse Brain*[23]. MassCube also reported a statistically significant decrease in histidine levels during aging. The observed decrease in histidine levels may reflect its reduced strength in mitigating oxidative stress in the brain[30]. Similarly, cellular senescence has been associated with decreased proline synthesis, which impairs mitochondrial function and contributes to neurodegeneration[31]. This aligns with MassCube's findings, further supporting the link between aging and diminished proline levels. MassCube also identified pronounced effects on aging-related metabolites in the mouse brain, such as CDP-ethanolamine and homoserine lactone, whose specific biological roles in the aging process remain to be uncovered. With the automatic strategy for selecting chemical discriminators behind the phenotype, MassCube facilitates efficient preliminary filtering of protentional biomarkers, prioritizing metabolites with reliable analytical performance and biological effects for downstream biological validation.

Using the 32 most discriminatory metabolites, global differences were revealed between adolescence and middle-aged mouse brain as shown in the UMAP plot (Fig. 5d). Further evaluation revealed an AUC of 1.000 in the ROC curve, with a fivefold cross-validation accuracy of $0.991 \pm 0.008$ (Fig. 5e), indicating the power of MassCube's phenotype classification module.

## Discussion

MassCube addresses long-standing challenge in metabolomics, starting with nontargeted peak detection. Other software typically uses 'rate-of-change' methods, determining the presence and boundaries of peaks by calculating derivatives of intensity changes of masses from scan to scan, or data-dependent extracted ion chromatograms (EICs)[7,21,32]. MassCube considers first principles, specifically how S/N, peak resolution and the intensity ratio of coeluting ions influence peak detection. At its core, MassCube balances the trade-off between sensitivity and robustness during peak detection. External validated datasets for rigorous benchmarking are still missing. Such datasets would present large-scale ground-truth data generated by different LC-MS methods and spanning biological matrices, with careful validation performed by a group of expert analytical chemists. To bridge this data gap, we used both synthetic and experimental data to develop the software, including 27,000 simulated peaks in addition to 722 randomly selected experimental peaks to test the performance of MassCube against classic MS-DIAL, MZmine and *xcms* software. This strategy to employ two types of benchmarking datasets has not been used before in metabolomics software development but may add confidence that MassCube will perform robustly for various LC-MS methods in different laboratories. However, convoluted experimental data also display characteristics that are difficult to explore in a systematic way using synthetic data, such as varying scan numbers across peaks, peak tailing, baseline dips, sudden peak spikes, ion saturation, and different types of background noise. The benchmarked software we compared MassCube against use different algorithms. MS-DIAL calculates the first and second derivatives of mass bins, MZmine3's ADAP algorithm[32] detects peaks from data-dependent EICs, and *xcms* applies data-dependent EICs to a wavelet transform for peak detection. However, our benchmarking tests showed false positive and false negative peak detections remained a substantial challenge for these software programs, because nontargeted peak detection is faced with poor peak shapes, high noise levels, and partially resolved peaks. Human-defined rules for peak shapes, reflected in algorithmic parameters, cannot account for all such scenarios. Hence, rate-of-change approaches may simply be unable to robustly address such problems in experimental datasets. Instead, MassCube focuses on MS signal clustering and peak edge detection through local minimum identification. Because MassCube does not rely on the rate-of-change for peak findings, it showed consistently superior performance to benchmarked software in synthetic benchmarking data, including tests across different levels of noise added to the signals. This finding was true for both double-peaks or single-peaks (Fig. 2e). Furthermore, MS-DIAL, MZmine3, and *xcms* performed even worse when benchmarked against complex experimental data. In contrast, MassCube maintained its accuracy against 722 experimental peaks, consistent with the results from synthetic data, achieving an average accuracy of over 95% for all tested peaks. Unlike other software, MassCube showed efficient handling of thousands of reported peaks, including peak metadata reports (such as asymmetry factor, Gaussian similarity, and noise score) that can be used for subsequent data curation. While other software tools often provide signal-to-noise information, comprehensive peak quality metadata is not comprehensively implemented in MS-DIAL, MZmine3, and *xcms*, sometimes even displaying erroneous reports.

In addition, parameter settings are crucial for software performance. In benchmarking, we deliberately avoided parameter tuning for any software across all test datasets. Instead, we used default, recommended settings to prevent overfitting and ensure fair comparisons. While some software might perform better with study- or matrix-specific adjustments, we argue that reliance on such tuning indicates lower robustness compared to alternatives that perform well across diverse conditions.

Besides accuracy, MassCube's peak detection has high efficiency along with low memory cost. The fast computation time benefits from (a) eliminating computational overhead that is associated with dynamic peak modeling in classic rate-of-change peak detection, (b) optimized Python array programming, which is much faster than tools built on less efficient programming environments such as R, and (c) parallel computing capabilities that enable MassCube to scale efficiently for large datasets. Over the past decades, several integrated MS data processing tools have been developed in R, including XCMS, RforMassSpectrometry, and TidyMass. Using the *xcms* R package, we demonstrated that R exhibits lower data handling efficiency than Python, which limits its application in large-scale metabolomics research and ultra-high throughput MS processing, such as Orbitrap Astral data. In proteomics, Python-based software has been developed like pyOpenMS[33] and Dinosaur[34]. In metabolomics, the Python-based *asari* software[35] is superior in feature detection compared to *xcms*, but it lacks comprehensive functionalities beyond MS1 data processing. Although low-level languages like C could offer even faster performance, we developed MassCube in Python to balance high efficiency with ease of use and readability. For instances requiring intensive optimization, compiled languages such as Cython and Numba can be employed to further enhance MassCube's efficiency.

MassCube covers the full workflow for metabolomics and lipidomics. Using the *Metabolome Atlas of the Aging Mouse Brain* data, we demonstrated several core modules in MassCube, including direct annotation of ISFs and multimodal MS/MS identification that are not available in MS-DIAL, MZmine3, or *xcms*. MassCube utilizes multiple criteria for ISF annotation, including the presence of fragments in the MS/MS spectrum of the parent ion and the similarity of peak shapes. Consequently, MassCube provides a conservative approach to ISF annotation: if the parent ion is not detected due to complete fragmentation in the ion source, its corresponding ISFs cannot be automatically annotated using this approach.

While MassCube does not directly perform metabolite identification, it enhances annotation accuracy in several ways. Metabolite annotation depends on accurate and comprehensive peak detection, in addition to robust in-source fragment and adduct annotation. MassCube advances the state of the art of in-source fragment and adduct annotation, thereby improving overall compound annotation accuracy. By reducing the number of false negative peak reports compared to benchmarked software, MassCube provides more robust data for statistical analyses to ensure that more biologically relevant features are used for downstream annotation.

Data processing in MassCube requires no coding skills and can be easily performed by analytical chemists, validated during the software beta testing. While a graphical user interface (GUI) is not available in the current version, we explicitly designed MassCube with non-programming users in mind by providing a series of command-line applications for direct usage (Supplementary Fig. 1), along with step-by-step instructions on our project website (https://huaxuyu.github.io/masscubedocs/docs/quickstart/). MassCube also offers comprehensive visualization tools and automatic figure generation capabilities for in-depth data inspection. MassCube also employs direct classification tools to distinguish groups defined in the experimental design. With the advanced ecosystem of machine learning in Python, MassCube enables prediction of experimental classes with preliminarily selected chemical discriminators. The MassCube's phenotype classifier builder also implements immediate validation after model construction, ensuring model accuracy and robustness. Although the selected metabolites demonstrate reliable analytical performance based on peak metadata and show distinct concentration levels across the queried phenotypes, downstream biological or clinical experiments remain necessary to confirm their biological function and validate the generated hypotheses. Moreover, while MassCube

provides reliable metabolomics data reports, confident biological insights would benefit from other software or databases explicitly design for data interpretation or integration of data from different assays or studies as well.

MassCube has been tested on Windows, macOS, and Linux systems ranging from personal laptops to high-performance computing clusters. The framework with MS data objects supports the seamless integration of advanced algorithms, making it adaptable to community contributions. MassCube could also be used as a backend for cloud computing and user-interfaced applications. To support data sharing and re-analysis, the built-in metadata tracking system ensures standardized records of each data processing task with parameters. Active development is ongoing for MassCube, aiming to provide users with additional tools for data processing such as time-series analyses, SERRF data normalization, formula predictions, chemical classifications and providing a GUI.

In the next decade, MS data processing will face challenges stemming from new configurations of instrumentation. While small metabolomics projects with fewer than 100 samples do not face major computational resource constraints, the increasing complexity and scale of data processing may present challenges in the future. Mass spectrometers like Orbitrap Astral MS are already pushing the boundaries of data acquisition by parallelizing mass analyzers, a trend that is likely to continue in future developments. These advancements will introduce new concepts and data structures, necessitating the extension of conventional data objects as well as the creation of flexible and scalable computational tools. MassCube is designed to meet these demands by offering the flexibility needed to accommodate future MS data types.

As the scale of community MS data collection continues to grow, researchers are increasingly eager to integrate information from different studies for broader discoveries. With databases like GNPS, MetabolomicsWorkbench, and MetaboLights, repository-level analysis at the MS/MS level is now possible. However, a substantial amount of information is often lost when chromatograms are not fully utilized. We believe that with its advanced infrastructure and efficiency, MassCube is poised to uncover hidden insights from the vast sea of public data, leading to a deeper understanding of instrumentation, small molecules, and biology.

## Methods

### Ethical statement

The Alzheimer's disease dataset used for speed benchmarking is an exploratory study coordinated by Duke University under R. Kaddurah-Daouk. The protocol was approved by the UC San Diego's Institutional Review Board (IRB) protocol #202063, Indiana University IRB study #1011003338, Kansas University IRB study #CR00020412, University of Wisconsin IRB study IORG0000056 approved 3-29-2023, New York University IRB study #i20-00427. Written informed consent was obtained from all participants.

### Animal studies

As described in the initial study, mice were cohoused by gender groups of 4–5 in individually ventilated cages (Optimice IVC, Animal Care Systems, Centennial, CO) on a 12:12-h (6:00/18:00) light:dark cycle at 68–79 °F with 40–60% humidity and provided water and standard rodent chow (Rodent chow, Harlan 2918) ad libitum. Brain tissue samples were collected from 3, 16, 59, and 92 weeks old male and female wild-type mice on a C57BL/6 N background. All procedures were approved by the IACUC of the University of California, Davis, which is an AAALAC-accredited institution. Animal housing and euthanasia were performed in accordance with the recommendations of the Guide for the Care and Use of Laboratory Animals.

**MassCube project.** MassCube is a Python computing library with a total of 16 modules to handle a wide range of data processing tasks and fundamental objects designed for LC-MS/MS data, including raw data management, parameters, feature detection, feature evaluation, feature grouping, normalization, annotation, alignment, network analysis, statistical analysis, visualization, phenotype classifier builder, workflows and other utility functions. Documentation for MassCube is available at https://huaxuyu.github.io/masscubedocs.

**Raw MS data parsing and metadata management.** MassCube supports standardized raw MS data in mzML and mzXML formats. Raw MS data are read and parsed into the *MSData* object, which was designed to organize high-dimensional LC-MS/MS data, manage metadata, and facilitate downstream processing and visualization. Within the *MSData* object, scans are organized sequentially, with each scan represented as a *Scan* object. MassCube automatically records the timestamp of each individual file for quality control, intensity normalization, and batch effect correction. Metadata for data processing were recorded and stored as a Python *dictionary* object including assembly of modules, data processing parameters, and versions of dependencies. The metadata are automatically exported with timestamp for trackable and repeatable analysis. Examples were shown in Supplementary Note 1.

**Feature detection.** MassCube feature detection clusters MS signals from unique ions, followed by Gaussian filter-assisted feature segmentation. A region of interest (ROI) is defined as an $m/z$ value that appears continuously across multiple MS1 scans. Let there be a total of $n$ MS1 scans denoted as $S = s_1, s_2, ..., s_n$. The search begins with the first scan $s_1$, initializing a set of ROIs using all $m/z$ values in $s_1$. Suppose there are $m_1$ $m/z$ values in $s_1$, then we have $m_1$ ROIs initiated from $s_1$, denoted as $R = ROI_1, ..., ROI_{m1}$. The algorithm then proceeds to the second scan $s_2$, matching the $m/z$ values in $s_2$ to the initiated ROIs. If a match occurs within a given $m/z$ tolerance (default = 0.01 Da), the ROI will be extended. Otherwise, a gap is detected. Upon completion of each new MS1 scan, the algorithm transfers all ROIs with gap numbers exceeding a specified threshold (default = 30) to the final set of ROIs, which will no longer be extended in subsequent calculations. This process iterates through all MS1 scans up to $s_n$. Subsequently, a feature segmentation algorithm was used to separate different ion species within a given ROI. For $ROI_i$, suppose it contains $k$ MS1 scans, resulting in $k$ intensity values represented as an array $I = int_1, ..., int_k$. First, a one-dimensional Gaussian filter is applied to smooth the signal array $I$, generating the smoothed signal $I'$ for edge detection. Subsequently, peaks above the baseline are captured based on peak prominence ratio calculated using the *SciPy* Python package. The local minimum between any two adjacent peaks are determined as the edges for segmentation. Importantly, using the scan number of detected edges, the original signal array $I$ rather the smoothed array $I'$ is segmented, with corresponding intensities reported by MassCube. Two parameters are critical for a successful peak segmentation that is robust to noise and can accurately separate two ion species. One parameter is the sigma ($\sigma$) that controls the level of smoothing in the Gaussian filter. Over-smoothing the signal results in loss of peak shape details and inability to distinguish coeluted isobaric species, while under-smoothing makes the signal too sensitive to noise, leading to incorrect segmentation of a single peak. The other parameter is the prominence ratio, which measures how much a peak stands out from the surrounding baseline and is defined as the ratio of vertical distance between the peak and its lowest contour line to ($I'$).

**Feature grouping.** MassCube systematically computes the correlations between detected features for annotating isotopes, adducts, and in-source fragments (ISFs). The algorithm considers the $m/z$ difference, retention times, scan-to-scan correlations calculated by Pearson's product-moment correlation coefficient, and MS/MS spectra. It

sequentially annotates isotopes, ISFs, and adducts. This implies that features annotated as isotopes will not be considered as ISFs or adducts, and features annotated as ISFs will not be considered as alternative adducts.

Annotation of isotopes is critical for compound annotation, determination of charge states, and determination of the presence of halogen atoms. MassCube annotates isotopes based on the $m/z$ difference and retention times. Charge state is further determined based on isotope patterns. By default, singly and doubly charged ion species are considered, and users may define the range of charge states to be considered in MassCube. ISF is a major source of false feature annotations. MassCube annotates ISFs based on $m/z$, retention times, scan-to-scan correlations, and MS/MS spectra. An annotated ISF must meet the following criteria: (1) its precursor $m/z$ presences in the parent MS/MS spectrum; (2) its scan-to-scan correlation with the parent ion at commonly detected scans is higher than tolerance (default = 0.7). When the number of common scans is not sufficient (less than 5) to calculate scan-to-scan correlation, retention time is matched by default tolerance = 0.05 min. MassCube annotates adducts based on $m/z$, retention time, and scan-to-scan correlation. By default, $[M + H]^+$, $[M + H-H_2O]^+$, $[M+Na]^+$, $[M + K]^+$, $[M + NH_4]^+$, $[2M + H]^+$, $[3M + H]^+$, $[M + 2H]^{2+}$ are considered in positive ion mode; $[M-H]^-$, $[M-H-H_2O]^-$, $[M+Cl]^-$, $[M + CH_3OO]^-$, $[M + HCOO]^-$, $[2M-H]^-$, $[3M-H]^-$, $[M-2H]^{2-}$ are considered in negative ion mode. Users may choose to consider additional adduct types or define custom adduct types in MassCube. A pair of annotated adducts must meet the following criteria: (1) their $m/z$ difference agrees with the defined value; (2) their scan-to-scan correlation is higher than tolerance (default = 0.7). When the number of common scans is not sufficient (less than 5) to calculate scan-to-scan correlation, retention time is matched by default tolerance = 0.05 min.

**Feature evaluation.** Reporting chromatographic peak metadata is essential for researchers to focus on Gaussian-shaped peaks and avoid noise. Three metrics are automatically calculated and reported including asymmetry factor, Gaussian similarity, and noise score. More metrics can be easily computed using the *masscube* Python package, including the variance of $m/z$ values and retention time, peak width and the average intensity of the top three scans (i.e., top average). Asymmetry factor measures how symmetrical a peak is by comparing the distances from the center to its two flanks. For a given chromatographic peak with $n$ MS1 scan points, let the set of MS signal intensities be $S = s_1, s_2, ..., s_n$, corresponding to $n$ retention times $T = t_1, t_2, ..., t_n$. The retention time of the apex $t_a$ is defined as

$$t_a = t_{argmax(S)} \tag{1}$$

MassCube then identifies the first MS1 scan point from the apex $t_a$ to the left where the intensity falls below 10% of the peak height, denoted as $t_{left}$, and similarly to the right, yielding $t_{right}$. The asymmetry factor is then calculated as:

$$asymmetry\ factor = \frac{t_{right} - t_a}{t_a - t_{left}} \tag{2}$$

If $t_{left} = t_a$, the asymmetry factor is defined as 99. This convention is used to indicate a scenario where the left side of the peak is the apex, suggesting a highly asymmetrical peak.

Gaussian similarity quantifies the resemblance of a given chromatographic peak to a perfect Gaussian function. The algorithm first fits a Gaussian function to the data $S = s_1, s_2, ..., s_n$, resulting in the fitted signal $S' = s_1', s_2', ..., s_n'$. The cosine similarity score is then computed between the original signal and the fitted signal:

$$Gaussian\ similarity = cosine(S, S') \tag{3}$$

This score ranges from −1 to 1, where a score closer to 1 indicates a high similarity to a Gaussian function. The noise score measures the fluctuation of the MS signal over time. For a given feature data set $S = s_1, s_2, ..., s_n$, a data point $s_i$ is considered a turning point if it meets the following condition:

$$(s_i - s_{i-1}) \times (s_{i+1} - s_i) < 0, i \in 2, ..., n-1 \tag{4}$$

The total number of turning points is counted as $p$, and the noise score is calculated as:

$$noise\ level = \frac{p - 1}{n - 2} \tag{5}$$

When a peak is perfectly smooth, the noise score is 0, as there is only one turning point at the apex.

**Standard-free retention time correction.** MassCube requires no additional input, such as a list of internal standards or known metabolites including $m/z$ and reference retention time for retention time correction. The algorithm is designed to be applicable even in the absence of spiked internal standards. The anchor features for retention time correction are selected using QC samples. Suppose $n$ QC samples are analyzed. MassCube first selects the QC sample $QC_i$ with the highest total intensity as the reference. Subsequently, features are sorted by $m/z$ values from low to high, leading to $F_1, ..., F_m$. All features are examined sequentially. Let $mz_j$ and $noise_j$ refer to the $m/z$ value and noise score of $F_j$, respectively. A feature $F_j$ is considered a valid retention time anchor if the following conditions are met:

$$\left(mz_j - mz_{j-1} > mz_{tol}\right) \wedge \left(mz_{j+1} - mz_j > mz_{tol}\right) \wedge (noise_j < noise_{tol}) \tag{6}$$

where $mz_{tol} = 0.01$ and $noise_{tol} = 0.3$ by default. Finally, the valid retention time anchors with the top 50 peak heights are further selected and split into training and testing data. For a given sample, the retention times of the selected anchors are located, and outliers are removed to prevent failed retention time correction. A linear interpolation model is then established for retention time correction.

Evaluation of the retention time correction algorithm was performed using the mouse brain dataset conducted separately for four ion modes. The evaluation workflow is as follows: (1) Load the retention time correction models, which were automatically exported after MassCube data processing. (2) Obtain the retention times of the testing anchors for evaluation, which are different from the training features used for model construction. (3) Apply the retention time correction models to the retention times of the testing anchors. (4) Compare the corrected retention times to the reference retention times.

**Feature alignment and gap filling.** MassCube identifies the commonly detected features across different files for alignment using $m/z$ and retention time, with default tolerances set to 0.01 Da and 0.2 min, respectively, which are tunable. Gap filling is performed by forced peak-picking using raw MS data, where the highest MS signal is identified within a specified retention time window (default = reference retention time ± 0.05 min).

**Multimodal feature annotation.** MassCube annotates features by $m/z$-RT match, identity search and fuzzy search using ***Flash Entropy Search*** algorithm implemented in *ms-entropy* Python package (ver. 1.2.2). MS/MS spectral preprocessing is crucial for accurate spectral matching. In MassCube, preprocessing includes (a) precursor ion exclusion, (b)

noise filtering using both relative and absolute intensity thresholds (default: ions with relative intensity <1% of base peak intensity are removed; ions with absolute intensity <10,000 for Orbitrap MS and <500 for QTOF MS are removed), and (c) m/z range-restricted matching. For example, if MS data was collected from m/sz = 100–400, then MassCube only considers fragments from 100–400 for database matching. In practice, similarity scores between 0.7 and 0.8 are commonly used for putative annotations in metabolomics. A similarity score of 0.7 was chosen in MassCube by default based on its previously published entropy similarity search algorithm[36]. MassCube supports multiple MS/MS database formats including MSP, JSON, and pickle. Public MS/MS libraries for metabolomics and lipidomics, sourced from the MS-DIAL public MS/MS database, have been prepared in *pickle* format for faster loading speeds and are available at https://zenodo.org/records/11363475.

**Normalization.** MassCube offers two types of data normalization. The first type addresses differences in total sample amount or concentration, such as those found in urine samples. While Probabilistic Quotient Normalization algorithm is used in MassCube by default, the optimal normalization method can vary depending on the dataset and is often debated. We therefore provide alternative algorithms, such as summed intensity, for user selection. The second type focuses on correcting systematic MS signal drifts in large-scale data acquisition. To address this, MassCube includes QC-based normalization. Timestamps from raw data are automatically acquired and utilized to guide this normalization process, requiring no need for user input.

**Phenotype classifier builder and data visualization.** MassCube provides univariate and multivariate models for pairwise and multi-group comparisons. For more advanced, machine learning-based statistical analysis, MassCube includes the phenotype classifier builder. This module leverages the *scikit-learn* package to perform a range of tasks, including data scaling, univariate feature selection, model construction, cross-validation, and evaluation metrics such as ROC curves. Moreover, the visualization module in MassCube is available for the rapid generation of publication-quality figures, facilitating both manual inspection and data overview. Examples of these visualizations are provided in Supplementary Fig. 9.

**Feature detection benchmarking**
**MassCube command line applications.** Mass spectrometry data processing is highly application focused. MassCube is designed to support applications by integrating individual functions and modules into workflows. These pre-defined workflows simplify and accelerate data processing and analysis for specific applications. The MassCube untargeted metabolomics workflow addresses specific bottlenecks in mass spectrometry-based data processing. It integrates metadata curation, feature detection, evaluation, alignment, annotation, and statistical analysis to provide users with a comprehensive view of the data. The untargeted metabolomics workflow is implemented as a command line application. First-time users can generate results in just four steps: (1) install Python, (2) install MassCube with a single command, (3) organize raw data files in a project folder and (4) run "untargeted-metabolomics" in the folder. A test data set is provided at https://zenodo.org/records/15173232. A quick start user instruction is available at https://huaxuyu.github.io/masscubedocs/docs/quickstart.

**Software and parameters.** MassCube 1.0.22, MS-DIAL 4.90, MZmine 3.90, and *xcms* R package 4.0.0 were used for performance benchmarking. Python 3.11.7 and R 4.3.0 were used for running *masscube* Python package and *xcms* R package, respectively. Complete parameters and code for data processing were provided at https://zenodo.org/records/14159704. To ensure a fair comparison, common key parameters across the compared software were standardized as follows: MS1 intensity tolerance was set to 30,000 for Orbitrap and 1000 for QTOF; m/z tolerance for MS1 scans was 0.01 Da, and m/z tolerance for MS/MS scans was 0.015 Da. Specific key individual parameters in each software were set by default as recommended by the respective tools:

(1) MassCube: sigma in Gaussian filter: 1.2; peak prominence ratio: 0.1; scan-to-scan Pearson's correlation coefficient tolerance for feature grouping: 0.7.
(2) MS-DIAL: mass slice width: 0.05 Da; smoothing algorithm: linear weighted moving average; smoothing level: 3 scans; minimum peak width: 5 scans.
(3) MZmine: peak detection algorithm: ADAP; smoothing algorithm: Savitzky Golay algorithm; retention time smoothing: 5; chromatogram resolving: local minimum resolver; minimum consecutive scans: 5 scans.
(4) *xcms*: peak detection algorithm: *centWave*; peak width: 5 to 60 s; signal-to-noise threshold: 0; prefilter: 3 to 100.

**Synthetic MS data.** Two types of synthetic MS signals were generated to represent double peaks and single peaks, respectively. For double peaks, we prepared all combinations of the three peak parameters as shown in Fig. 2b including S/N, intensity ratio, and peak resolution. Different S/N were achieved by tuning the amplitude of Gaussian noise added to the normal distribution relative to the peak height (2, 4, 6, 8, 10%). For double peaks, the intensity ratio represented the peak height ratio of the two peaks (1, 2, 3, 4, 5), and peak resolution was varied (1.00, 1.25, 1.50, 1.75, 2.00). Double peaks where the smaller peak had an S/N lower than 3 were excluded, as they fell below the limit of detection (LOD). For each combination, 1000 random replicates were generated for optimizing parameters in MassCube, resulting in a total of 110,000 single peaks and 110,000 double peaks. For algorithm performance comparison, we generated a synthetic mzML file to ensure that all software tools process the same file for a fair evaluation. This allows for a direct comparison of feature detection accuracy and efficiency across MassCube, MS-DIAL, MZmine3, and *xcms* under identical conditions. A new set of 100 replicates for each combination was generated with an additional condition for noise score = 0, yielding a total of 13,500 double peaks and 13,500 single peaks. All generated signals were inserted into raw QTOF MS data of human urine samples, with distinct m/z values outside the original mass range to avoid overlapping with the real signals. The list of inserted peaks, Python code for simulation, and the generated mzML data file are available at https://zenodo.org/records/14159704.

**Benchmarking with synthetic MS data.** MassCube 1.0.22, MS-DIAL 4.9, MZmine 3.90, and *xcms* R package 4.0.0 were all used to process the same synthetic mzML data file as described above. To ensure a fair comparison, the default parameters for QTOF were used across all four software platforms, as the original data were acquired from a QTOF MS. It is important to note that the settings for mass tolerance and intensity tolerance did not affect the results since the inserted synthetic MS signals were of high intensity (on the order of $10^6$) and exhibited no mass variation between different scans. This design ensures the focus on benchmarking the performance of different algorithms with respect to varying chromatographic peak shapes rather than other factors such as m/z variance.

**Benchmarking with experimental MS data.** Eight experimental LC-MS/MS data were used for benchmarking feature detection performance, including NIST SRM 1950 plasma data, acquired on Thermo Orbitrap Exploris 480 MS, reverse phase (RP) positive ion mode; NIST SRM 1950 plasma data, acquired on Thermo Q

Article

Exactive MS, RP negative ion mode; mouse plasma data, acquired on Thermo Q Exactive HF MS, hydrophilic interaction chromatography (HILIC) positive ion mode; NIST Human Fecal Material RGTM 10162 data, acquired on Thermo Q Exactive MS, HILIC negative ion mode; Human serum data, acquired on Bruker impact II QTOF MS, RP positive ion mode; Mouse feces data, acquired on Bruker impact II QTOF MS, HILIC positive ion mode; whole fruit fly (*D. melanogaster* strains) data, acquired on Bruker impact II QTOF MS, HILIC positive ion mode; and human urine data, acquired on Bruker impact II QTOF MS, HILIC negative ion mode. Raw data files, complete parameters and code for data processing, along with the output feature tables from all four tested software tools, are provided at https://zenodo.org/records/14159704. A total of 722 local extracted ion chromatograms (EICs) were randomly selected from the Orbitrap and QTOF data and manually labeled by experienced analytical chemists. The PNG files for labeled EICs, along with a detailed list of their corresponding *m/z* values and retention times, and information on whether they were identified by different software tools, are provided at https://zenodo.org/records/14159704. Speed benchmarking was performed using four metabolomics studies obtained from the MassIVE and MetaboLights repositories (see "**Data availability**"). Single-threading was used during the speed benchmarking of the four metabolomics studies on a Windows Desktop with an AMD Ryzen Threadripper PRO 2945WX 12-Core CPU and 32 GB of memory. Parallel processing was enabled for processing Orbitrap Astral MS data collected from human plasma samples from Alzheimer's Disease patients on a MacBook M3 Pro laptop with 36 GB of memory. Detailed sample preparation workflow and LC-MS experimental settings for analyzing the human plasma samples were listed in Supplementary Note 2. No biological metadata, including sex, was used in this study, because the dataset was solely intended for speed benchmarking and software stability demonstration.

**Biological application.** The raw MS data files were obtained from a previous study, acquired using a ThermoFisher Q-Exactive HF with a HESI-II ion source (Thermo Scientific, Waltham, MA, USA) coupled with a Vanquish UHPLC system (Thermo Scientific, Waltham, MA, USA). The study examined cohorts of 8 male and 8 female wild-type mice at four life stages: adolescence (AD, 3 weeks), early adulthood (EA, 16 weeks), middle age (MA, 59 weeks), and old age (OA, 92 weeks). Immediately following euthanasia, brains were harvested and dissected into 10 anatomically defined regions: cerebral cortex (CT), olfactory bulb (OB), hippocampus (HC), hypothalamus (HT), basal ganglia (BG), thalamus (TL), midbrain (MB), pons (PO), medulla (MD), and cerebellum (CB). In total, 640 brain samples were analyzed using two complementary assays including untargeted metabolomics via hydrophilic interaction chromatography (HILIC) and untargeted lipidomics via reversed-phase liquid chromatography (RPLC). Sample preparation and detailed experimental configurations are provided in Supplementary Note 3. Data analysis was performed using MassCube version 1.0.16. Data processing parameters and the exported feature table are available at Zenodo with accession code 14159704 [https://zenodo.org/records/14159704]. An outlier detection algorithm was developed in MassCube to automatically identify failed LC-MS injections, ensuring a clean data report and quality control. Details of the outlier detection algorithm are specified in Supplementary Note 4, with examples of detected outliers provided in Supplementary Fig. 10. UMAP visualization was conducted in Python using the *umap* package. Chemical classification was performed using the ClassyFire Batch Compound Classification tool (https://cfb.fiehnlab.ucdavis.edu). Chemical Similarity Enrichment Analysis was conducted using the ChemRICH tool (https://chemrich.fiehnlab.ucdavis.edu/).

**Reporting summary**

Further information on research design is available in the Nature Portfolio Reporting Summary linked to this article.

## Data availability

For software accuracy benchmarking: synthetic MS data and eight raw experimental LC-MS data files used for algorithm benchmarking are available at Zenodo with accession code 14159704. For speed benchmarking, orbitrap dataset #1 of NIST Human Fecal Material Standards was obtained from MassIVE with accession code MSV000086988 [https://massive.ucsd.edu/ProteoSAFe/dataset.jsp?task=68d120fefcf243dcb83cdf5f448c31a7]; orbitrap dataset #2 of urine metabolomics storage study was obtained from MassIVE with accession code MSV000091929 [https://massive.ucsd.edu/ProteoSAFe/dataset.jsp?task=25663ex4d7cc7410cbb0324b08c2c892a]; QTOF dataset #1 of plant metabolomics was obtained from MetaboLights with accession code MTBLS188; QTOF dataset #2 of Type 1 diabetes plasma lipidomics study was obtained from MetaboLights with accession code MTBLS620. For demonstration of peak evaluation criteria, 41 public datasets used for modeling the distribution of peak metadata were obtained from MetaboLights, with their study identifiers provided at Zenodo with accession code 14159704. For biological application, data of the *Atlas of the Aging Mouse Brain* can be accessed from the Metabolomics Workbench under Project ID PR001047. Data from the NIST human fecal material standards are available from the MassIVE repository under MSV000086989 [https://massive.ucsd.edu/ProteoSAFe/dataset.jsp?task=2f73277b4e034948acebfdf1edab17ed]. For metabolite annotation, the NIST23 Tandem Mass Spectral Library used in biological applications can be purchased from NIST or other distributors [https://www.nist.gov/programs-projects/nist23-updates-nist-tandem-and-electron-ionization-spectral-libraries]; the licensed NIST23 Mass Spectral Library can only be accessed after purchase, as per NIST policy. Supplementary Data Files including raw output from MassCube by re-analyzing the data of *the Atlas of the Aging Mouse Brain*, eight experimental LC-MS data files used for algorithm benchmarking, EICs in.png format for manually labeled experimental data for algorithm benchmarking, MassCube's ouput of 41 metabolomics studies including 200 individual files from MetaboLights and MassCube's output of the phenotype classifier for mouse brain data are available at Zenodo with accession code 14159704. The human plasma data of Alzheimer's Disease patients collected on Orbitrap Astral MS are available on MassIVE with accession code MSV000097583 [https://massive.ucsd.edu/ProteoSAFe/dataset.jsp?task=100416b91eb24735a53eb2eecf2fd3d6]. Unless otherwise stated, all data supporting the results of this study can be found in the article, supplementary, and source data files. Source data are provided with this paper.

## Code availability

All the source code for the MassCube project is available on GitHub [https://github.com/huaxuyu/masscube] with https://doi.org/10.5281/zenodo.15151320 [https://doi.org/10.5281/zenodo.15151320]. Python and R code for data processing are available at Zenodo with accession code 14159704. Outline for source code is provided in Supplementary Note 5.

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

## Acknowledgements

This work was funded by USDA Agricultural Research Service 2021-67017-35783 (O.F.), NIH R01 GM155383 (O.F.), and NIH U01 AG08862 (O.F.). Samples from the National Centralized Repository for Alzheimer's Disease and Related Dementias (NCRAD), which receives government support under a cooperative agreement grant (U24 AG021886) awarded by the National Institute on Aging (NIA), were used in this study. We thank contributors who collected samples used in this study, as well as patients and their families, whose help and participation made this work possible. Samples are contributed by the NIA-funded ADRCs: P30 AG072976 (PI Andrew Saykin, PsyD); P30 AG066512 (PI Thomas Wisniewski, MD); P30 AG062429 (PI James Brewer, MD, PhD); P30 AG062715 (PI Sanjay Asthana, MD, FRCP); P30 AG072973 (PI Russell Swerdlow, MD).

## Author contributions

H.Y. and O.F. designed the research. H.Y. developed the MassCube Python package, prepared documentation, established the data simulation algorithm, prepared benchmarking datasets, performed algorithm benchmarking, and data processing for biological application. J.D. acquired the *Metabolome Atlas of the Aging Mouse Brain* dataset and reviewed the data reprocessing results. T.S. acquired the Orbitrap Astral MS dataset and reviewed the data reprocessing results. M.L. performed software testing as analytical chemist without guidance. Y.L. supported implementation of the entropy search algorithm into MassCube. H.Y. and O.F. wrote the manuscript with contributions from all other authors.

## Competing interests

The authors declare no competing interests.
