## [Transparent Peer Review file · Nature Communications]

MassCube improves accuracy for metabolomics data processing from raw files to phenotype classifiers

Corresponding Author: Professor Oliver Fiehn

Version 0:

Reviewer comments:

Reviewer #1

(Remarks to the Author)

The manuscript introduces MassCube, a Python-based framework for metabolomics data processing, highlighting its advantages over existing tools like MS-DIAL, MZmine3, and xcms. The framework's ability to achieve 100% signal coverage with robust reporting of chromatographic metadata is commendable, as is its efficiency in handling large MS data sets. The successful application of MassCube to the Metabolome Atlas of the Aging Mouse Brain data, despite batch effects, demonstrates its potential for advancing omics and biomedical research. Overall, the work is highly relevant and addresses critical challenges in the field of metabolomics, representing a significant advancement in metabolomics data processing. However, there are areas where the manuscript could be improved to enhance clarity, rigor, and impact.

Specific Comments:

1. The manuscript could better emphasize the feature of identifying in-source fragments to eliminate interference in MassCube. While the use of D5-glutamine and D5-glutamic acid in Figure 4 is a good illustration, demonstrating real metabolites would enhance the validation section. Exploring if natural glutamine and glutamic acid exhibit similar in-source fragments, and identifying common characteristics among the 2,604 detected in-source fragments in the mouse dataset would strengthen the manuscript. Further validation and discussion of these aspects would showcase MassCube's capability in identifying in-source fragments more comprehensively.
2. The impact of in-source fragment correction on the final results should be addressed. Clarifying how correcting for in-source fragments affects the status of metabolites like glutamine in batch samples as differential metabolites would enhance the interpretation of MassCube's results.
3. While MassCube appears adept at distinguishing segments from noise or isomeric peaks, showcasing its ability to discover new isomeric metabolites or identify isomeric metabolites as potential biomarkers would enhance its scientific understanding and impact.
4. Experimental evidence demonstrating MassCube's ability to mitigate batch effects is lacking.
5. Consider restructuring the Results section into distinct subsections like "Feature Detection," "Benchmarking," "Experimental Validation," and "Biological Application" for improved readability and coherence.
6. Please provide additional details on the benchmarking methodology, including the criteria for selecting synthetic and experimental datasets, and clarify the rationale behind parameter selection for each software tool.
7. Please provide a more detailed discussion on the strengths and weaknesses of MS-DIAL, MZmine3, and xcms relative to MassCube to better explain why MassCube outperforms in certain areas.
8. Enhance the interpretability of figures, such as using more intuitive color gradients and ensuring comprehensive legends that allow figures to stand alone.
9. Ensure all code, including scripts used for benchmarking and data processing, are well-documented on GitHub for

enhanced reproducibility.

10. Discuss the broader implications of applying MassCube to the Metabolome Atlas of the Aging Mouse Brain and its alignment with existing literature and potential applications.

11. Include a brief discussion on potential future developments for MassCube, such as adaptation to other omics data types or integration with AI-based drug discovery tools.

12. Define acronyms at first use, maintain terminology consistency, and proofread for grammatical errors and typos.

13. Provide a toy example for users to quickly familiarize themselves with MassCube's functionality and ensure the accessibility of the quick start guide provided by the authors.

(Remarks on code availability)

Reviewer #2

(Remarks to the Author)

This manuscript introduces MassCube, a Python-based framework for LC-MS metabolomics data processing. The authors highlight improvements in speed, accuracy, and comprehensive processing workflows compared to existing software, including MS-DIAL, XCMS, and MZmine. While MassCube appears to offer incremental advancements, several key issues need further consideration.

Comments

The authors claim superior performance of MassCube against other software through standardized benchmarking. However, these types of subjective evaluations raise concerns about fairness. It is easy to demonstrate superior results if competitor tools are not optimally configured, this has been a standard approach used to “validate” superiority when introducing new software. The authors themselves often presented the vast superiority of MS-DIAL for years.

While the reported speed and accuracy improvements can be beneficial, they appear incremental rather than groundbreaking. Most existing tools already perform well for many metabolomics applications. An improvement in speed or accuracy may not justify publication in a high-impact journal unless it addresses a critical bottleneck or introduces fundamentally new methodologies.

The authors emphasize MassCube's ability to handle large datasets efficiently. However, in metabolomics workflows, where significant resources and time (often many months) are spent on sample preparation and data acquisition, relatively small gains in data processing speed have limited practical impact. Users prioritize reliability, robustness, and biological insights over minor efficiency improvements.

The most critical aspect of metabolomics analysis—metabolite identification—is not sufficiently addressed in the manuscript. While the authors discuss in-source fragment annotation and fuzzy search approaches, these do not represent significant advancements in the field. Accurate metabolite identification remains a major challenge in LC-MS-based and LC-MS/MS-based metabolomics, and the paper does not offer solutions that significantly advance this area.

The manuscript could benefit from clearer explanations of how synthetic datasets were generated for benchmarking. This would help readers assess the robustness of the comparisons.

While the modular and Python-based architecture of MassCube is an advantage, the lack of a graphical user interface (GUI) may limit adoption by non-programming users.

MassCube aims to demonstrate improvements in speed and accuracy, but these appear incremental rather than transformative. The manuscript would benefit from addressing the fairness of benchmarking comparisons, providing more impactful advancements in metabolite.

(Remarks on code availability)

Reviewer #3

(Remarks to the Author)

The authors of this manuscript have created the masscube package for Python for full processing of chromatography-mass spectrometry data. The package includes all processing steps: peak detection, noise removal, peak matching between different chromatograms by retention time, library search and sample classification. This is an important task and such software may be in demand by the scientific community. There are many similar packages available, but the authors provide a quantitative comparison of their package with those of other authors and show that the package they have developed is significantly superior to those previously published. The authors describe the operation of their package (algorithms) in sufficient detail, and provide a fairly detailed description of all procedures, in particular, how they have worked with other software packages. All data are publicly available. The github repository looks good, there are installation instructions, the

package is available via pip. In my opinion the manuscript is well written, contains all the necessary information and can be published in a peer-reviewed scientific journal after minor revision. The novelty of this paper is not outstanding: many similar publications have been done in the past, but since this kind of software may be in demand in practice, I can hope that this paper will be useful.

1) It would be good to add some quantitative benchmark of RT drift correction to the paper. The authors describe the algorithm in quite a bit of detail, but how good is it compared to other packages? Would it be possible to add a benchmark (comparison with other programs) on synthetic or real data?

2) The authors write about mass spectral search, but library search quite often leads to wrong answer (and always if the right answer is not available in the database). How is the similarity score threshold chosen? The authors write "a flash entropy search similarity cutoff of 0.7" - why 0.7? The authors write "This fuzzy search extended the number of chemically classified compounds 2-3 fold across the four assays". How reliable are these classifications and annotations? The authors write as a merit of their approach that they have annotated many features, but do not provide any assessment of the reliability of these annotations. Is it possible to give a quantitative assessment?

3) The section "feature annotation" in Methods is written very briefly, no quantitative parameters are available. The authors should expand this section significantly. Are they using their implementation of algorithms for database search?

4) Are there plans to develop this project further? Unfortunately, very often software stops developing after the publication of an article in a journal. Is it possible to add some mention about further plans? Also I recommend to make (at least for Windows) a release with built-in python. For example with the package github.com/astral-sh/python-build-standalone. So that it can be downloaded and run offline without any installation and without internet access. This would be easier for many users, and less prone to deprecation of required dependency versions. But this does not apply to the manuscript per se.

5) The authors mention speed and low RAM requirements as the main advantages of their approach. In my opinion it's worth cutting these statements down: at the current level of technology development it's not a big problem.

In general, the manuscript is relatively high-quality and after minimal revision can be published.

(Remarks on code availability)

The project is successfully installed and can be launched. The repository contains documentation. I recommend adding an example of the data on which workflow works. The code is quite good.

Version 1:

Reviewer comments:

Reviewer #1

(Remarks to the Author)

I have thoroughly reviewed the revised manuscript along with the detailed responses to the reviewers' comments. I am pleased to note that the authors have addressed all of my feedback and suggestions. Consequently, I have no further concerns with this latest version.

(Remarks on code availability)

Reviewer #2

(Remarks to the Author)

While the authors present a well-documented and benchmarked software tool, MassCube does not introduce a fundamentally new methodology or significantly advance the field of metabolomics beyond existing software solutions. The improvements in speed and accuracy, while useful, are largely incremental, and the lack of a robust approach to metabolite identification limits its broader impact. Additionally, the reliance on similarity-based annotation raises concerns about the reliability of reported metabolites. While MassCube may be a useful tool for some applications, its contributions do not rise to the level of a significant advance in the field.

(Remarks on code availability)

It is usable

Reviewer #3

(Remarks to the Author)

The authors have made corrections to their manuscript as suggested by the reviewer. In addition, the authors have attached test data on which their software package works. Despite the fact that the scientific novelty of the work done is quite limited, I think the manuscript can be considered for publication in its current form, since the developed software can be used in practice.

(Remarks on code availability)

The software can be installed and works. If one uses the test data attached by the authors, the program's work leads to a result. The only thing is that I could recommend deleting the hidden service files of the MacOS system from the archive. The files are not needed for work. Also at my Linux system the software prints many many times message "font family Arial not found". Despite this plots look fine. The source code is clear and at first glance looks good.

Reviewer #1 (Remarks to the Author):

The manuscript introduces MassCube, a Python-based framework for metabolomics data processing, highlighting its advantages over existing tools like MS-DIAL, MZmine3, and xcms. The framework's ability to achieve 100% signal coverage with robust reporting of chromatographic metadata is commendable, as is its efficiency in handling large MS data sets. The successful application of MassCube to the Metabolome Atlas of the Aging Mouse Brain data, despite batch effects, demonstrates its potential for advancing omics and biomedical research. Overall, the work is highly relevant and addresses critical challenges in the field of metabolomics, representing a significant advancement in metabolomics data processing. However, there are areas where the manuscript could be improved to enhance clarity, rigor, and impact.

Answer: We sincerely appreciate the reviewer's constructive feedback and positive assessment of our work. We have carefully revised the manuscript to improve its clarity, rigor, and impact. We believe these revisions have strengthened the manuscript and are grateful for the reviewer's input.

Specific Comments:

1. The manuscript could better emphasize the feature of identifying in-source fragments to eliminate interference in MassCube. While the use of D5-glutamine and D5-glutamic acid in Figure 4 is a good illustration, demonstrating real metabolites would enhance the validation section. Exploring if natural glutamine and glutamic acid exhibit similar in-source fragments and identifying common characteristics among the 2,604 detected in-source fragments in the mouse dataset would strengthen the manuscript. Further validation and discussion of these aspects would showcase MassCube's capability in identifying in-source fragments more comprehensively.

Answer: We have followed this suggestion and now exemplify how to annotate in-source fragments using endogenous (unlabeled) glutamine and glutamic acid. Additionally, we now better clarify that MassCube annotates in-source fragments by (a) calculating the scan-to-scan correlations with a larger-mass putative parent ion and (b) by detecting the precursor ion of the likely in-source fragment in the MS/MS spectrum of the putative parent ion. Due to the scan-to-scan correlation, and the presence of the precursor ion in the MS/MS spectrum of the putative parent, MassCube yields high probability for correctly associating in-source fragments, as long as the parent molecule is actually detectable (and generates a MS/MS spectrum in the data acquisition methods that users apply). Hence, MassCube currently groups features and reports likely in-source fragments, but we do not automatically combine such features. We now added that for the 2,604 detected in-source fragments, 7.5% (194 out of 2,604) were classic water losses, and 4.1% (107 out of 2,604) were losses of ammonia. Future investigation may detail further classes of in-source fragmentations for different matrices and mass spectrometers. We also clarify that the detection of in-source fragmentation fails if no parent (intact) molecule is detected.

Comment

2. The impact of in-source fragment correction on the final results should be addressed. Clarifying how correcting for in-source fragments affects the status of metabolites like glutamine in batch samples as differential metabolites would enhance the interpretation of MassCube's results.

Answer: MassCube currently labels likely in-source fragment features, but does not automatically combine this data with parent molecule abundances. So, intensities of metabolites are not affected. However, we now clarify that indeed, the detection of in-source fragments may add confidence to the annotation of parent ions in a similar way as the detection of multiple adducts (such as $[2M+H]^+$ or $[M+K]^+$) already add confidence, if their MS/MS spectra are found in MS/MS libraries. Moreover, the annotation of features as in-source fragments helps in avoiding false metabolite annotations. For example, cholesterol esters often cause in-source fragments of "cholesterol". Associating such false "cholesterol" annotations as in-source degradation of larger cholesterol esters helps reducing false discovery rates. We now clarify this novel addition of automatic detection of in-source fragmentation to data processing software, but we also state that this process only works if also the parent (intact) molecule is present.

Comment

3. While MassCube appears adept at distinguishing segments from noise or isomeric peaks, showcasing its ability to discover new isomeric metabolites or identify isomeric metabolites as potential biomarkers would enhance its scientific understanding and impact.

Answer: We now add this idea into the discussion section as one possibility how MassCube can be used. However, our test dataset was not designed to detect specific biomarkers for disease phenotypes. Often, isomers are co-regulated by biological phenomena and may not be found as differential biomarkers, unless such isomers are genuine positional isomers. We now searched for lipidomic pairs of isomers with respect to aging, brain or regional differences and found examples that we added to the **Figure 4a-b**.

Comment

4. Experimental evidence demonstrating MassCube's ability to mitigate batch effects is lacking.

Answer: Before, we demonstrated the normalized MS signal drift including batch effect of an endogenous metabolite as an example in **Supplementary Figure 1**. Additionally, the PCA plot illustrated reduced data variance after LOWESS normalization. We now have expanded this validation by comparing improvements in reproducibility. By using 80% QC samples as training data and leaving out 20% of QC samples as testing data, we showed the reduced data variation in four assays including HILIC positive, HILIC negative, RP positive and RP negative ion modes (**Supplementary Figure 1**). We now add a comment that further improvements could be achieved by adding SERRF to the code.

Comment

5. Consider restructuring the Results section into distinct subsections like "Feature Detection," "Benchmarking," "Experimental Validation," and "Biological Application" for improved readability and coherence.

Answer: We have now restructured the manuscript into “Software functionality”, “Benchmarking using synthetic MS data”, “Benchmarking using experimental MS data”, “Biological application by data re-analysis”, “Biological application by machine learning-based phenotype classification”.

Comment

6. Please provide additional details on the benchmarking methodology, including the criteria for selecting synthetic and experimental datasets, and clarify the rationale behind parameter selection for each software tool.

Answer: We have now added arguments for using these specific benchmarking tests. We are the first to use synthetic mzML-based datasets to test software, so that (a) all software is tested on identical input data and (b) ground truth values are known in the test. We also now clarify that the 13,500 true single peaks and 13,500 true double peaks resulted from the permutation of peak parameters including peak shapes and noise levels by varying signal-to-noise ratio, intensity ratio, and peak resolution.

We also add rationale for the choice of experimental data, we now better explain why we chose a matrix of immediate biomedical relevance (plasma, human and mouse) as well as a matrix of immense complexity (fecal matter), i.e. *matrix independence*. For that reason, we used NIST standard reference materials, to enable other researchers to re-use the data or employ additional tests with the same samples on their own instruments (*reusability*). We also clarified why we chose different dataset, to showcase that the software works on both small-scale and large-scale studies (*size independence*), in addition to testing both Orbitrap and QTOF datasets (*instrument independence*).

In terms of parameter settings in software comparisons, we now clarify that we consciously and intentionally did not perform hyperparameter optimizations. Instead, all software was tested under identical and consistent parameters to avoid parameter overfitting, and to ensure that fair testing was performed. Key parameters were standardized (MS1 intensity, m/z tolerance in MS1 and MS/MS), whereas software-specific parameters were set to their default values as recommended by each tool. In the discussion section, we added a comment that it is possible that other software might better perform if software-specific parameters were adjusted for each matrix, or each study (matrix/instrument combination). However, we argue that software should be regarded as poor-performing (in comparison to more robust, alternative software), if outcomes relied on adjusting parameters for each study and each matrix. That is the main reason why we not only performed benchmark tests for in-silico synthetic data, but also experimental data with different matrices.

Comment

7. Please provide a more detailed discussion on the strengths and weaknesses of MS-DIAL, MZmine3, and xcms relative to MassCube to better explain why MassCube outperforms in certain areas.

Answer: We have further expanded our discussion on why MassCube outperforms other software in terms of accuracy and efficiency. (1) MassCube does not rely on the “rate of change” in chromatographic peak detection. Instead, its design to consider “regions of interest” ensures that MassCube is robust to noise and signal fluctuations. (2) For “regions of interest”, MassCube reports

all consecutive mass signals across MS1 scans, regardless of peak shape. Hence, MassCube ensures comprehensive data reporting without relying on variable peak-shape definitions. (3) MassCube evaluates peak Gaussian similarity, noise levels, and asymmetry factors. Users can filter and focus on peaks based on robust metadata, reducing the likelihood of statistical analysis being affected by background noise. (4) Finally, MassCube achieves high computational efficiency through three key factors. (4a) Eliminating computational overhead that is associated with dynamic peak modeling in classic rate-of-change peak detection. (4b) Optimized Python array programming, which is much faster than tools built on less efficient programming environments such as R. (4c) Parallel computing capabilities, allowing MassCube to scale efficiently for large datasets. We summarize these arguments in the discussion now.

Comment

8. Enhance the interpretability of figures, such as using more intuitive color gradients and ensuring comprehensive legends that allow figures to stand alone.

Answer: We have improved the information content of the figure legends to ensure that each Figure can be understood and interpreted by itself. We have reviewed the clarity of color gradients in Figures but we have to concede that the impact of color use is highly subjective and that no general guideline can be given. We have asked all co-authors on input to get representative feedback, and incorporated their comments for improved clarity.

Comment

9. Ensure all code, including scripts used for benchmarking and data processing, are well-documented on GitHub for enhanced reproducibility.

Answer: All code scripts have been documented with appropriate notes. The benchmarking and data processing scripts are shared through Zenodo (<https://zenodo.org/records/14159704>) to ensure proper versioning and accessibility (original manuscript lines 718-719 and code availability). The GitHub repository is specifically intended to host the MassCube Python package and track its development. We have now added an outline for code use in the **Supplementary Note 4**.

Comment

10. Discuss the broader implications of applying MassCube to the Metabolome Atlas of the Aging Mouse Brain and its alignment with existing literature and potential applications.

Answer: We have now added statements on the impact of our findings with respect to the 'Aging Mouse Brain' atlas, including other aging studies.

Comment

11. Include a brief discussion on potential future developments for MassCube, such as adaptation to other omics data types or integration with AI-based drug discovery tool.

Answer: We added comments for future developments for MassCube in the discussion section, including for SERRF normalization, joining adducts for quantification, and improved use of internal

standards for quantification. We continue to outline our future development plans on our website (<https://huaxuyu.github.io/masscubedocs/docs/plans>), ensuring transparency for users and encouraging collaboration with researchers interested in contributing to the project, and we now give this URL in the Supplementary Information.

Comment

12. Define acronyms at first use, maintain terminology consistency, and proofread for grammatical errors and typos.

Answer: We checked all acronyms and terminology.

Comment

13. Provide a toy example for users to quickly familiarize themselves with MassCube's functionality and ensure the accessibility of the quick start guide provided by the authors.

Answer: We have now prepared a test data set (<https://zenodo.org/records/14869318>) that is useful for a quick start for users. A quick start instruction was also provided at <https://huaxuyu.github.io/masscubedocs/docs/quickstart/>.

Reviewer #2 (Remarks to the Author):

This manuscript introduces MassCube, a Python-based framework for LC-MS metabolomics data processing. The authors highlight improvements in speed, accuracy, and comprehensive processing workflows compared to existing software, including MS-DIAL, XCMS, and MZmine. While MassCube appears to offer incremental advancements, several key issues need further consideration.

Answer: We appreciate the recognition of MassCube's improvements in speed, accuracy, and comprehensive processing workflows. We do not agree that such improvements are incremental, as we will highlight below.

Comment

The authors claim superior performance of MassCube against other software through standardized benchmarking. However, these types of subjective evaluations raise concerns about fairness. It is easy to demonstrate superior results if competitor tools are not optimally configured, this has been a standard approach used to "validate" superiority when introducing new software. The authors themselves often presented the vast superiority of MS-DIAL for years.

Answer: Fairness is the most importance aspect we considered during benchmarking. Benchmarking is the number #1 most important step in comparing outcomes and improvements. Unfortunately, benchmarking is not always recommended or highlighted by authors or by reviewers. Here, we do the opposite: we not only claim MassCube to be superior, we validate this opinion.

Hence, we have taken extensive measures to ensure that our comparisons are as objective and reproducible as possible. To achieve this, we implemented the following strategies:

1. **Standardized Key Parameters:** We ensured that all software tools were benchmarked using the same key parameters, including intensity cutoff and mass tolerance, to maintain consistency across evaluations (see “software and parameters” in Methods section)
2. **Use of Default Parameters:** Since different algorithms operate under distinct mathematical assumptions, we used default parameters for each software tool, as these are generally optimized by the developers for broad applicability (see “software and parameters” in Methods section).
3. **Diverse Data Selection:** To prevent bias or overfitting towards MassCube's parameters, we used a diverse range of datasets, including both synthetic and experimental data. Specifically, we included eight experimental LC-MS/MS datasets from Thermo Fisher Orbitrap Q Exactive and Bruker QTOF Impact II instruments. These datasets covered both metabolomics and lipidomics analyses of NIST SRM 1950 human plasma, human serum, mouse plasma, NIST RGTM 10162 fecal matter, whole-body fruit flies, and human urine samples (see Data Availability section)

Additionally, to ensure clarity and reproducibility, we have detailed all software versions and package dependencies used during benchmarking, allowing independent verification of our results. We appreciate the reviewer's concerns and believe that these measures reinforce the objectivity and fairness of our comparative analysis. We have now clarified our rationales for software benchmarking in the revised Results section.

Comment

While the reported speed and accuracy improvements can be beneficial, they appear incremental rather than groundbreaking. Most existing tools already perform well for many metabolomics applications. An improvement in speed or accuracy may not justify publication in a high-impact journal unless it addresses a critical bottleneck or introduces fundamentally new methodologies.

Answer: We acknowledge the reviewer's concerns. However, software accuracy is a critical benchmark item, and relying on inferior software is bad for science. We do not agree that existing tools already perform well: in metabolomics, the crisis in reproducibility and comparability between labs and across studies is widely debated! MassCube presents a major step forward in accuracy (as we have shown, see also below), and transparency. For double peak (isomer detection), MassCube has >93% accuracy, while our former software MS-DIAL only has 63% accuracy and the MZmine 3.9 has only 30% accuracy. We revisited how we presented this important data, and have now revised Figure 3 with a simpler color scheme that better highlights MassCube's improvements in comparison to all other software. Secondly, speed matters! We invite the reviewer to try and use another software, ThermoFisher's CompoundDiscoverer. We left it out of our comparison, because it is a commercial software. Nevertheless, it is so slow that it can only be used for very small sample sets.

1. Accuracy matters

The improvement in accuracy addresses a major bottleneck in metabolomics. Our benchmarking results for 722 ion trace peaks demonstrate that MassCube massively, and significantly, outperforms other leading software.

	Accuracy in double-peak detection	Accuracy in single-peak detection
MassCube	93.5%	97.3%
MS-DIAL	63.0%	89.8%

MZmine3	30.4%	81.8%
XCMS	28.3%	70.3%

Lack of accuracy is generating poor data. Poor data generates poor science. If the improvements were only very minor, like 1% improvement, we would agree: then, perhaps, such improvements should not go to a high-profile journal. But these improvements matter. They matter because a major challenge in metabolomics is the reliable resolution of isomeric peaks. Existing software struggles, leading to inaccurate quantification and compromised biological interpretation. MassCube's improved peak detection ensures more precise feature extraction, reducing errors in downstream analyses.

2. Speed Matters

MassCube's improvements enable large-scale studies that were previously impractical. MassCube achieves 6.5 times faster feature detection than xcms, 4 times faster than MS-DIAL, and 1.2 times faster than MZmine3, which is already optimized for speed. As mass spectrometry (MS) and metabolomics technologies advance, studies are increasingly generating larger datasets with more samples. This growing data complexity demands faster and more scalable software solutions, which MassCube effectively provides.

3. Efficiency matters.

Memory efficiency eliminates computational barriers for next-generation MS data (and for users who have small workstations). On a MacBook M3 Pro (36 GB RAM, 12 cores), MassCube processed a large-scale dataset in just 64 minutes, with peak memory usage of only 10.4 GB. This demonstrates that MassCube can efficiently handle next-generation MS data, which is eight times larger than traditional Orbitrap datasets, without requiring high-performance computing infrastructure. Memory efficiency is a critical limitation of existing tools, often preventing large datasets from being processed on standard hardware. MassCube removes this barrier, making high-throughput metabolomics accessible even on standard computing systems.

By addressing these key bottlenecks in accuracy, speed, and memory efficiency, MassCube is not merely an incremental improvement—it removes fundamental computational limitations that have hindered large-scale and next-generation metabolomics studies. We believe these advancements justify the high impact of the manuscript. We have now highlighted the importance for a leading software in metabolomics to provide both accuracy and speed. Our comprehensive benchmarking datasets strongly support that MassCube has achieved this standard.

Comment

The authors emphasize MassCube's ability to handle large datasets efficiently. However, in metabolomics workflows, where significant resources and time (often many months) are spent on sample preparation and data acquisition, relatively small gains in data processing speed have limited practical impact. Users prioritize reliability, robustness, and biological insights over minor efficiency improvements.

Answer: We appreciate the reviewer's perspective. It is correct that there are studies with small datasets, and studies with large datasets. However, we do not agree with the reviewer's notion that small studies would require many months on sample preparation and data acquisition. Typical data

acquisition run times today are always <30 min, often even <10 min. Small studies with <50 samples are therefore typically acquired in one day per assay.

Hence, the reviewer's argument does not hold. Metabolomics workflows only need months for sample preparation and data acquisition if either very many samples are processed, or if new data acquisition methods need to be developed. MassCube is not only more accurate (see above), but also faster, enabling large-scale metabolomics studies that involve thousands of samples. Existing tools often face long processing times and high memory consumption, making analysis impractical on standard desktop or laptop.

Beyond speed, MassCube ensures reliability and robustness through superior peak detection accuracy (97.3% for single peaks, 93.5% for double peaks). Simply put, other software is less reliable, and less robust. MassCube is more reliable, and more robust. MassCube enables comprehensive metadata reporting, allowing users to filter unreliable features and improve biological insights. Low RAM usage (10.4 GB for 636 Thermo Astral Orbitrap files) enables processing on a standard laptop, making high-throughput metabolomics more accessible. Thus, we believe MassCube is not just about minor efficiency gains but removes computational barriers that limit large-scale and next-generation metabolomics research.

The reviewer states that users would also prioritize software to reveal biological insights. We agree, but this is harder to prove for any type of software. While MassCube focuses on reliability and robustness, we also showcase how better data leads to better biological interpretation, using a previously published dataset (on the Aging Mouse Brain Atlas). Obviously, biological insights would benefit from other types of software (and other types of data, or data integration) as well. MassCube focuses on data processing more than on biological data interpretation. We now added a statement to this effect into the conclusions.

Comment

The most critical aspect of metabolomics analysis—metabolite identification—is not sufficiently addressed in the manuscript. While the authors discuss in-source fragment annotation and fuzzy search approaches, these do not represent significant advancements in the field. Accurate metabolite identification remains a major challenge in LC-MS-based and LC-MS/MS-based metabolomics, and the paper does not offer solutions that significantly advance this area.

Answer: Our manuscript does not directly address metabolite identification, and neither did we state this objective. Nevertheless, metabolite annotation relies on accurate and comprehensive peak detection, as well as robust in-source fragment and adduct annotation. MassCube improves the state of the art for in-source fragment and adduct annotation, and hence, helps to improve compound annotations. Moreover, if a feature is not detected in the first place, it cannot be annotated or reported. Compared to classic software, MassCube demonstrated significantly higher peak detection accuracy on both synthetic and experimental benchmarking datasets, ensuring that more biologically relevant features are captured for downstream annotation.

Additionally, false annotation of in-source fragments or adducts as unique metabolites is a common issue in existing data processing software. MassCube addresses this challenge using a scan-to-scan correlation-based in-source fragment annotation strategy, reducing false positives and improving annotation reliability. To address the reviewer's concerns, we have now clarified the factors contributing to false metabolite annotation and have highlighted the strategies implemented in MassCube in the revised discussion section.

Comment

The manuscript could benefit from clearer explanations of how synthetic datasets were generated for benchmarking. This would help readers assess the robustness of the comparisons.

Answer: We sincerely appreciate the reviewer's feedback on our benchmarking strategies. In the Methods section, we have expanded the description of how the synthetic datasets were generated to improve clarity. Additionally, we have provided URL for synthetic dataset and an outline in **Supplementary Note 4** to clarify corresponding Python script and generated raw mzML files, ensuring transparency and reproducibility.

Comment

While the modular and Python-based architecture of MassCube is an advantage, the lack of a graphical user interface (GUI) may limit adoption by non-programming users.

Answer: We explicitly designed MassCube with non-programming users in mind by providing a series of command-line applications for direct usage, along with step-by-step instructions on our project website (<https://huaxuyu.github.io/masscubedocs/docs/quickstart/>). First-time users can generate results in just four steps: (1) install Python, (2) install MassCube with a single command, (3) organize raw data files in a project folder and (4) run "untargeted-metabolomics" in the folder. We have highlighted how non-programmer researchers better use MassCube in the revised discussion and the "MassCube command line applications" in the Methods section.

We also appreciate the reviewer's suggestion regarding GUI development. However, building a high-performance front-end interface needs both time and money. We recognize its value and plan to develop a GUI version of MassCube in the future. We have added this comment in the discussion.

Comment

MassCube aims to demonstrate improvements in speed and accuracy, but these appear incremental rather than transformative. The manuscript would benefit from addressing the fairness of benchmarking comparisons, providing more impactful advancements in metabolite.

Answer: We do not agree that our improvements in accuracy are incremental, as given above in the section "**Accuracy matters**". We have now made this improvement in accuracy clearer. We recognize the reviewer's concerns regarding benchmarking fairness and have clarified our approach to ensure fair comparisons, transparency, and reproducibility. This includes consistent parameter selection, diverse benchmarking datasets, and the use of the latest software versions. We have incorporated these clarifications in the revised manuscript to address the reviewer's concerns.

Reviewer #3 (Remarks to the Author):

Comment

The authors of this manuscript have created the masscube package for Python for full processing of chromatography-mass spectrometry data. The package includes all processing steps: peak detection, noise removal, peak matching between different chromatograms by retention time, library search and sample classification. This is an important task and such software may be in demand by the scientific community. There are many similar packages available, but the authors provide a quantitative comparison of their package with those of other authors and show that the package they have developed is significantly superior to those previously published. The authors describe the operation of their package (algorithms) in sufficient detail, and provide a fairly detailed description of all procedures, in particular, how they have worked with other software packages. All data are publicly available. The github repository looks good, there are installation instructions, the package is available via pip. In my opinion the manuscript is well written, contains all the necessary information and can be published in a peer-reviewed scientific journal after minor revision. The novelty of this paper is not outstanding: many similar publications have been done in the past, but since this kind of software may be in demand in practice, I can hope that this paper will be useful.

Answer: We thank the reviewer for their positive assessment of our manuscript and for recognizing the practical significance of MassCube for the scientific community. We appreciate the acknowledgment of our quantitative comparisons, detailed methodology, and open data availability. We have carefully addressed the revisions suggested to further enhance the manuscript.

Comment

1) It would be good to add some quantitative benchmark of RT drift correction to the paper. The authors describe the algorithm in quite a bit of detail, but how good is it compared to other packages? Would it be possible to add a benchmark (comparison with other programs) on synthetic or real data?

Answer: We appreciate the reviewer's insightful question. While retention time (RT) correction is included in MassCube as a supplementary function rather than a primary focus, we agree that validation is important. In original manuscript, we validated the retention time correction algorithm using the Aging Mouse Brain Atlas dataset. As shown in **Extended Data Fig. 3d**, we quantitatively validated the RT shift corrections using 32-38 independent additional RT testing anchors across four metabolomic assays.

Comment

2) The authors write about mass spectral search, but library search quite often leads to wrong answer (and always if the right answer is not available in the database). How is the similarity score threshold chosen? The authors write "a flash entropy search similarity cutoff of 0.7" - why 0.7? The authors write "This fuzzy search extended the number of chemically classified compounds 2-3 fold across the four assays". How reliable are these classifications and annotations? The authors write as a merit of their approach that they have annotated many features, but do not provide any assessment of the reliability of these annotations. Is it possible to give a quantitative assessment?

Answer: We appreciate the reviewer's concerns regarding spectral matching accuracy and database coverage. It is important to emphasize that similarity scores in metabolomics do not directly correspond to annotation accuracy, as the relationship between spectral matching scores and expected false discovery rates (FDRs) remains an open question in small molecule annotation.

In practice, similarity scores between 0.7 and 0.8 are commonly used for putative annotations in metabolomics. A similarity score of 0.7 was chosen in MassCube based on its previously published entropy similarity search algorithm, which has been rigorously benchmarked across multiple MS/MS databases. Manual verification of annotations is essential for biological studies to ensure data reporting confidence.

Fuzzy search expands the usage of MS/MS spectra and provide users with structural hypothesis of unknown metabolites. Further experiment is always required to elucidate their chemical structure. We have highlighted this point the revised manuscript to avoid misleading. MassCube's annotation algorithm is implemented from a previously validated entropy similarity search, where quantitative benchmarking is provided. To ensure transparency, we have highlighted this point in the revised Methods part (section: multimodal feature annotation).

Comment

3) The section "feature annotation" in Methods is written very briefly, no quantitative parameters are available. The authors should expand this section significantly. Are they using their implementation of algorithms for database search?

Answer: Each feature (m/z + retention time pair) may yield many MS/MS spectra per study. In MassCube, we use the best-matching MS/MS spectrum (acquired within a study) for compound annotations, and we specify this aspect now in the Method section. Feature annotation in MassCube is based on the previously published entropy similarity search and its improved version, flash entropy search. The `ms_entropy` Python package from flash entropy search has been integrated into MassCube for ultrafast spectral matching. The functionality of this approach has been rigorously benchmarked in a prior study (<https://www.nature.com/articles/s41592-023-02012-9>), and we now added additional comments to these two previously published papers. Additionally, we emphasize that MS/MS spectral preprocessing is crucial for accurate spectral matching. In MassCube, preprocessing includes (a) precursor ion exclusion, (b) noise filtering using both relative and absolute intensity thresholds, and (c) m/z range-restricted matching (for example, if MS data was collected from $m/z = 100-400$, then only fragment in database from 100-400 will be considered). To ensure clarity, we have now expanded the "Feature Annotation" section in the Methods section, providing quantitative parameters and further details on the database search process.

Comment

4) Are there plans to develop this project further? Unfortunately, very often software stops developing after the publication of an article in a journal. Is it possible to add some mention about further plans? Also I recommend to make (at least for Windows) a release with built-in python. For example with the package github.com/astral-sh/python-build-standalone. So that it can be downloaded and run offline without any installation and without internet access. This would be easier for many users, and less

prone to deprecation of required dependency versions. But this does not apply to the manuscript per se.

Answer: We greatly appreciate the reviewer's concern regarding the long-term development of MassCube. First of all, our code is publicly available, ensuring that other users or developers can use and adapt or improve it. Second, we are committed to actively maintaining and updating MassCube, continuously integrating advanced metabolomics data processing tools to enhance its functionality. We have done such improvements in the past, e.g. for MS-DIAL (now beyond vs. 4.9), but also for retip (now in retip2.0).

We added comments for future developments for MassCube in the discussion section, including advanced normalization algorithms like SERRF (PMCID: PMC9652764) or SERDA (PMCID: PMC10456436), joining adducts for quantification, and improved use of internal standards for quantification. We continue to outline our future development plans on our website (<https://huaxuyu.github.io/masscubedocs/docs/plans>), ensuring transparency for users and encouraging collaboration with researchers interested in contributing to the project, and we now give this URL in the Supplementary Information.

We appreciate the reviewer's comments on developing MassCube as a pre-built Python package. However, it is important to note that a standalone distribution may not be the most user-friendly option for non-programming users, as indicated in the official documentation of astral-sh/python-build-standalone (<https://gregoryszorc.com/docs/python-build-standalone/main/running.html>). In comparison, the current version of MassCube is designed for ease of use, requiring only the installation of Python and the MassCube package. Running data analysis is fully independent of an internet connection and requires just a single command: `untargeted-metabolomics`. Additionally, all dependencies are clearly listed in MassCube's Python package project file, ensuring they are automatically checked and updated during installation or updates.

To highlight MassCube's usability, we have added clarifications in the revised manuscript, emphasizing its straightforward installation and execution compared to standalone pre-built Python distributions.

Comment

5) The authors mention speed and low RAM requirements as the main advantages of their approach. In my opinion it's worth cutting these statements down: at the current level of technology development it's not a big problem.

Answer: We appreciate the reviewer's feedback and have added this statement in the discussion section. While computational resources continue to advance, we aim to highlight that even in current metabolomics studies, processing large-scale datasets (>1,000 files) on a standard PC or personal laptop remains a significant challenge for conventional software. Issues such as software crashes and excessively long processing times are common limitations. MassCube addresses these challenges by providing an efficient and scalable solution, ensuring practical usability for researchers working with large datasets.

In general, the manuscript is relatively high-quality and after minimal revision can be published.

Reviewer #3 (Remarks on code availability):

Comment

The project is successfully installed and can be launched. The repository contains documentation. I recommend adding an example of the data on which workflow works. The code is quite good.

Answer: We sincerely thank the reviewer for the positive assessment of our manuscript and code. We appreciate the suggestion to include an example dataset for demonstrating the workflow. To improve usability and reproducibility, we have now prepared a test data set (available at <https://zenodo.org/records/14869318>) that is useful for a quick start for users (instructions are available at <https://huaxuyu.github.io/masscubedocs/docs/quickstart/>).

REVIEWERS' COMMENTS

Reviewer #1 (Remarks to the Author):

I have thoroughly reviewed the revised manuscript along with the detailed responses to the reviewers' comments. I am pleased to note that the authors have addressed all of my feedback and suggestions. Consequently, I have no further concerns with this latest version.

Answer: We appreciate the reviewer's constructive comments and suggestions.

Reviewer #2 (Remarks to the Author):

While the authors present a well-documented and benchmarked software tool, MassCube does not introduce a fundamentally new methodology or significantly advance the field of metabolomics beyond existing software solutions. The improvements in speed and accuracy, while useful, are largely incremental, and the lack of a robust approach to metabolite identification limits its broader impact. Additionally, the reliance on similarity-based annotation raises concerns about the reliability of reported metabolites. While MassCube may be a useful tool for some applications, its contributions do not rise to the level of a significant advance in the field.

Answer: We appreciate the reviewer's positive comments regarding the software documentation and benchmarking. However, we disagree with the significance for the field. Accuracy in feature detection is critical for MS-based nontargeted metabolomics. True peak detections are the foundation for all subsequent steps, including adduct/in-source fragment grouping, quantification, feature annotation and statistical analysis. The accuracy in feature detection has not been systematically benchmarked before. Our work shows that MassCube outperforms the most important alternative software packages:

MassCube achieved **93.5% accuracy** for double peaks and **97.3% for single peaks**, compared to low double-peak detection accuracy: 63.0% for MS-DIAL 4.9, 30.4% for MZmine 3, and 28.3% for XCMS 4.0. We do not understand why the reviewer thinks that a **>30% improvement in accuracy** is just incremental. Instead, MassCube represents a significant advancement for the field.

We changed the title accordingly, to make the improvements in accuracy clearer.

Similarly, no other software provides **fuzzy similarity search** (sometimes called "molecular networking") for metabolite annotation. MassCube does that automatically. We do not understand why this inherent and fast computation is not a major progress for users? MassCube provides users 2–3 times more classified compounds compared to previous software methods (saving the users the need to upload their data to GNPS or MS-FINDER as secondary computation platforms).

Obviously, like for all software, manual interpretation of MS/MS data is still necessary to verify annotations and propose structures for further validation. In the future, we plan to develop and integrate more advanced spectral annotation strategies into

MassCube to further enhance its overall performance.

Reviewer #2 (Remarks on code availability):

It is usable

Answer: We appreciate the reviewer's efforts in testing the MassCube software.

Reviewer #3 (Remarks to the Author):

The authors have made corrections to their manuscript as suggested by the reviewer. In addition, the authors have attached test data on which their software package works. Despite the fact that the scientific novelty of the work done is quite limited, I think the manuscript can be considered for publication in its current form, since the developed software can be used in practice.

Answer: We thank the reviewer for the constructive comments. For novelty, please see our comments above. We changed the title accordingly, to make the improvements in accuracy clearer.

Reviewer #3 (Remarks on code availability):

The software can be installed and works. If one uses the test data attached by the authors, the program's work leads to a result. The only thing is that I could recommend deleting the hidden service files of the MacOS system from the archive. The files are not needed for work. Also at my Linux system the software prints many many times message "font family Arial not found". Despite this plots look fine. The source code is clear and at first glance looks good.

Answer: Hidden files from MacOS system have been removed. We have also updated MassCube now to avoid the font issue in Linux systems.